

# Do Drinfeld twists of $AdS_5 \times S^5$ survive light-cone quantization?

**Stijn J. van Tongeren**[*] **and Yannik Zimmermann**[†]

Institut für Physik, Humboldt-Universität zu Berlin
IRIS Gebäude, Zum Grossen Windkanal 6, 12489 Berlin, Germany

[*] svantongeren@physik.hu-berlin.de , [†] yannik.zimmermann@physik.hu-berlin.de

## Abstract

We study how a wide class of Abelian Yang-Baxter deformations of the $AdS_5 \times S^5$ string behave at the quantum level. These deformations are equivalent to TsT transformations and conjectured to be dual to beta, dipole, and noncommutative deformations of SYM. Classically they correspond to Drinfeld twists of the original theory. To verify this expectation at the quantum level we compute and match (1) the bosonic two-body tree-level worldsheet scattering matrix of these deformations in the uniform light-cone gauge, and (2) the Bethe equations of the equivalent model with twisted boundary conditions. We find that for a generalization of gamma deformations of the BMN string the we are able to express the S matrix either through a Drinfeld twist or a shift of momenta. For deformations of the GKP string around the null-cusp solution we encounter calculational obstacles that prevent us from calculating the scattering matrix.

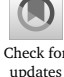

# 1  Introduction

In the difficult arena of exact results in quantum field theory and string theory the discovery and development of integrability has provided astonishing insights [1, 2]. One of the prime examples is the free string on $AdS_5 \times S^5$, receiving attention due to its role in the AdS/CFT correspondence. Taking this maximally supersymmetric sigma model as a starting point, less symmetric deformations that still preserve integrability attracted attention. Many of these can be phrased as Yang-Baxter (YB) deformations [3–5], see [6] for a recent review.

The YB deformations of the $AdS_5 \times S^5$ string have a multitude of different interpretations in terms of string theory and AdS/CFT. The so-called inhomogeneous deformations – building on solutions of the modified (or inhomogeneous) classical Yang-Baxter equation – lead to trigonometric quantum deformations of the symmetry algebra [7–9]. In contrast, the homogeneous deformations – based on solutions of the (homogeneous) classical Yang-Baxter equation (CYBE) – are expected to correspond to Drinfeld twists of the symmetry bialgebra. This expectation is due to a one-to-one correspondence between solutions to the CYBE and such Drinfeld twists [10, 11]. It is supported by a calculation of the classical monodromy matrix for various models [12, 13].

A subclass of homogeneous YB deformations are the Abelian deformations. Their quantum theory and its relation to the Drinfeld twist will be the focus of this paper. Abelian deformations give superstring models, i.e. they are Weyl-invariant [14]. Further, they are the most studied variety of YB deformations. Examples include the gravity dual of canonical noncommutative SYM [15–17], Dipole type backgrounds [18, 19], Schrödinger geometries [18, 20], and most famously the real $\beta$ deformation [21–23]. For this last deformation it was also shown that it can be equally well described by taking the undeformed model and introducing twisted boundary conditions for the string [22].

Coming to pure $AdS_5$ deformations, in [24] it was conjectured that Abelian deformations depending on the $AdS_5$ Cartan generators are dual to super Yang-Mills (SYM) theory on non-commutative spacetimes. The argument was later extended in [25] to arbitrary homogeneous YB deformations, guided by the expected Drinfeld-twisted symmetry of the deformations and the fact that noncommutative SYM shows the same behavior of its symmetry algebra.

Concretely, a Drinfeld twist of the symmetry algebra typically changes the S matrix of the model to

$$S \to F_{21} S F^{-1}, \tag{1.1}$$

where $F$ is the twist matrix that in the case of Abelian deformations with deformation matrix $r$ schematically takes the form

$$F = \exp(ir). \tag{1.2}$$

More detail will be presented in section 2.3. The central theme of this paper is to calculate the S matrix for a big class of Abelian deformations and check if the expected statement of the last two equations holds.

Studies of the quantum theory have been previously done for the three-parameter generalization of the real $\beta$ deformation in two different ways: Firstly, the authors of [26,27] approached the problem through the use of the undeformed model with twisted boundary conditions. They implemented these twisted boundary conditions through inserting an appropriate twist matrix into the transfer matrix. Secondly, the works [28,29] took the Drinfeld twist perspective and directly implemented one part of the three-parameter deformation at the level of the Lagrangian. This deformed Lagrangian was then used to calculate the perturbative T matrix; the result matched the tree-level term of the proposed Drinfeld twist (1.1) of the full S matrix [29]. The authors then used this twisted S matrix to construct the Bethe equations [28]. For this one-parameter deformation this gives the same result as the twisted boundary conditions construction of the first papers. In the present paper we want to use the same two angles to examine the quantum theory of various Abelian deformations. The choice of Abelian deformations we can treat this way is limited through a complication arising from gauge fixing.

Concrete quantization and S matrix calculations of the $AdS_5$ string require a gauge-fixing procedure. This inevitably interferes with the symmetry structure of the $AdS_5 \times S^5$ string: The full symmetry algebra is $\mathfrak{psu}(2,2|4)$; after light-cone gauge fixing, for example, only $\mathfrak{su}(2|2)_{ce}^{\oplus 2} \subset \mathfrak{psu}(2,2|4)$ is still realized linearly, for the most-used gauge choice of the BMN string. The problem is that it is only for a gauge-fixed model with a smaller linearly realized symmetry algebra, that an S matrix can be determined. Now, as argued above, upon deformation we expect a Drinfeld twist of the full algebra. It is however unclear how the gauge-fixing procedure interacts with the Drinfeld twist and in particular if the reduced S matrix inherits the twisted structure from the full theory. Hence, the central question of this paper is

*How does gauge fixing affect the expected Drinfeld twist of the S matrix?*

The one existing calculation of [29] shows that the twists survives for a particular one-parameter $\gamma$ deformation. However this particular deformation does not interact with the light-cone coordinates of the gauge-fixing, nor does it affect the $AdS_5$ part of the geometry. With this paper we want to lift these restrictions and answer the question for all possible Abelian deformations built from the six Cartan generators of the light-cone symmetry algebra $\mathfrak{su}(2|2)_{ce}^{\oplus 2}$.[1]

As was done in the two above mentioned works on $\gamma$ deformations we approach the quantum versions of the deformed models from two angles. Firstly, we directly compute the tree-level scattering matrix from the Yang-Baxter-deformed and gauge-fixed Lagrangian. Secondly, we use the fact that Abelian deformations are equivalently described by the undeformed model with twisted boundary conditions [12,22,33,34]. These can be implemented with a twist matrix in the Bethe equations, which then can be compared to the Bethe equations directly derived from the Drinfeld-twisted S matrix. In the end we will therefore have two independent checks of the deformed S matrices: a perturbative one of the tree-level term, and an all-loop one of the Bethe equations.

Our result for the form of the gauge-fixed S matrix is twofold. For most deformations we find the expected Drinfeld twist, while for a subset of deformations we get an S matrix with shifted momentum dependence.

**Drinfeld twist** The first case happens for all deformations with r matrices built from the shift generators for the $AdS_5$ isometry directions $\psi_{1,2}$, the $S^5$ isometry directions $\phi_{1,2}$, and

---

[1]The inhomogeneous $\eta$ deformation is also compatible with the BMN light-cone gauge, and moreover can be combined with the present Abelian deformations. The S matrices for various inhomogeneous deformations were studied in [7,9,30–32]. The S matrix for the combined deformation presumably readily follows from these results combined with our present ones.

the light-cone isometry direction $x^+$. For such a deformation matrix $r$ the gauge-fixed S matrix is then of Drinfeld-twisted form (1.1) with twist

$$F = \exp\left(\tfrac{i}{2h} r\right) \tag{1.3}$$

and string tension $h$. This S matrix matches both the perturbative T matrix of the deformed Lagrangian and the Bethe equations of the model with twisted boundary conditions. The shift generators for $\psi_{1,2}$ and $\phi_{1,2}$ act linearly on the fundamental excitations of the S matrix; while the shift generator for $x^+$ acts in the twist like the negative worldsheet Hamiltonian, $P_+ = -H_{ws}$, which matches the expectation from the calculation of the uniform light-cone gauge. Section 3.1.1 contains the full discussion of this case.

**Momentum shift** The second case occurs for all terms of the r matrix that include the generator $P_-$ of the light-cone isometry direction $x^-$. For example for an r matrix $r = P_- \wedge P_\nu$, where $P_\nu$ is one of the remaining five Cartan generators, the deformed S matrix takes the shifted form

$$S(p_1, p_2) = S^0(p_1 - P_\nu^1, p_2 - P_\nu^2), \tag{1.4}$$

where $S^0$ is the undeformed S matrix and $P_\nu^i$ acts on the $i$-th particle. It appears impossible to express this shift as a Drinfeld twist, as we will explain below. Again this S matrix matches both the perturbative T matrix of the deformed Lagrangian and the Bethe equations of the model with twisted boundary conditions. This behavior of deformations with generator $P_-$ does not match the expectation from the uniform light-cone gauge calculations. The full discussion can be found in section 3.1.2.

Besides these main results we discuss how the S matrix in the model with twisted boundary conditions needs to receive a correction term that is similar in nature to the momentum shift from above. It accounts for the twist in the boundary condition for the $x^+$ coordinate and originates from a necessary modification of the gauge-fixing prescription. More details are given when we discuss the uniform light-cone gauge in section 2.4 and the calculation for the model with twisted boundary conditions in section 3.2.

The second part of the paper looks at a second gauge-fixed model, the GKP string. While the BMN string has its light-cone directions used for gauge fixing lying in $AdS_5$ and $S^5$, the GKP string has its light-cone directions lying only in $AdS_5$. This choice leads to a different gauge-fixed theory, which was previously used to calculate the perturbative scattering matrix around its so called null-cusp solution [35, 36]. We investigated the behavior of Abelian YB deformations in this theory. However we encountered difficulties: both the perturbative treatment of the deformed model and the twisted boundary condition picture presented difficulties. Already in the undeformed case the expansion around the non-trivial null-cusp solution required a complicated field redefinition to eliminate an explicit worldsheet dependence of the Lagrangian. This construction does not survive the deformation of the model, leaving the deformed Lagrangian explicitly worldsheet-depended and further introducing worldsheet-dependence into the expressions for the shift generators; this prohibited us to determine any S matrices. Section 4 gives details.

In the next section we start with an introduction to various well-known concepts used throughout the paper, after which we will head into the actual calculations for the BMN string in section 3.

# 2 Prerequisites

Here we will present all basic definitions and derivations that are needed throughout the different sections of the paper. This includes Abelian Yang-Baxter deformations, Drinfeld twists

of symmetry algebras, the uniform light-cone gauge and the derivation of the Bethe equations through the transfer matrix.

## 2.1 Yang-Baxter deformed string on AdS$_5 \times$ S$^5$

Usually the string and its Yang-Baxter deformation are written in the coset formulation. We are only interested in the bosonic part of the theory, so we directly use the bosonic non-linear sigma-model formulation. The undeformed string action is

$$-\frac{h}{2} \int \mathrm{d}^2\sigma \left(\sqrt{-\det(h_{\gamma\delta})}h^{\alpha\beta} - \epsilon^{\alpha\beta}\right)G_{MN}\partial_\alpha x^M \partial_\beta x^N\,, \tag{2.1}$$

where $h$ is the string tension, $h_{\alpha\beta}$ and $\epsilon^{\alpha\beta}$ the worldsheet metric and epsilon tensor, and $G_{MN}$ and $x^M$ the AdS$_5 \times$ S$^5$ metric and coordinate fields.

The Yang-Baxter deformation was introduced through the insertion of an $R$ operator into the coset action [5,8]. Here we prefer to work directly at the level of the geometry, and introduce it through the two-tensor $r$ (that we also call r matrix) as follows

$$-\frac{h}{2} \int \mathrm{d}^2\sigma \left(\sqrt{-\det(h_{\gamma\delta})}h^{\alpha\beta} - \epsilon^{\alpha\beta}\right)(\tilde{G}_{MN} + \tilde{B}_{MN})\partial_\alpha x^M \partial_\beta x^N\,,$$
$$\text{with } \tilde{G} + \tilde{B} = \left(G^{-1} + r\right)^{-1}\,. \tag{2.2}$$

We represent $r$ on the tangent bundle through the Killing vector representation of $\mathfrak{psu}(2,2|4)$ on super AdS$_5 \times$S$^5$. The quantities $\tilde{G} + \tilde{B}$ describe the metric and B field of the deformed background. The two-tensor $r$ is antisymmetric and solves the homogeneous classical Yang-Baxter equation

$$[r_{12}, r_{13}] + [r_{12}, r_{23}] + [r_{13}, r_{23}] = 0\,. \tag{2.3}$$

## 2.2 Abelian Yang-Baxter deformations and related concepts

Out of the many possible solutions to the CYBE we want to focus on the simplest case, the Abelian solutions, generating Abelian YB deformations. They are defined as

$$r = \sum_{\mu\nu} \Gamma_{\mu\nu} A_\mu \wedge A_\nu \qquad\qquad A_\mu, A_\nu \in \mathfrak{psu}(2,2|4)\,,$$
$$= \sum_{\mu\nu} \Gamma_{\mu\nu}\left(A_\mu \otimes A_\nu - A_\nu \otimes A_\mu\right), \qquad [A_\mu, A_\nu] = 0\,. \tag{2.4}$$

One well known example of such an Abelian deformation is the real $\beta$ deformation corresponding to the Lunin-Maldacena background [21]; we will discuss its S matrix below. For this deformation the interesting observation was made that it can be equally well described by a TsT-transformed model [22].

**Relation to TsT transformations** This relation to TsT transformations is a general result for all Abelian YB deformations [34].[2] Assume $x_A$ and $x_B$ are target space coordinates whose shift symmetry gets generated by $A$ and $B$ respectively, i.e. $x_A \to x_A + \epsilon$ under the action of $\epsilon A$. Then an Abelian YB deformation with $r = \Gamma A \wedge B$ corresponds to a TsT transformation with steps

1. T dualize in $x_A$,

2. shift $x_B \to x_B + \Gamma \tilde{x}_A$,

3. T dualize in $\tilde{x}_A$,

where $\tilde{x}_A$ is the T dual of $x_A$. The general case of $r$ being a sum of terms is equivalent to a sequence of such TsT transformations.

---

[2]In general, homogeneous deformations are equivalent to non-abelian T duality transformations [37,38].

**Relation to twisted boundary conditions**   Another discovery through the example of the $\beta$-deformed string is that a YB deformation with again $r = \Gamma A \wedge B$ can be equally well described through the undeformed string with twisted boundary conditions [22][3]

$$
\begin{aligned}
x_A(R_\sigma) - x_A(0) &=: \Delta x_A = -\Gamma B\,, \\
x_B(R_\sigma) - x_B(0) &=: \Delta x_B = +\Gamma A\,,
\end{aligned}
\tag{2.5}
$$

where $R_\sigma$ is the circumference of the worldsheet cylinder. Here $A$ and $B$ are operators that read off the charges of the classical field configuration, or quantum state, under consideration. In other words, the boundary conditions are field configuration (state) dependent. We can generalize linearly to multi term $r = \sum_{\mu\nu} \Gamma_{\mu\nu} A_\mu \wedge A_\nu$ through

$$
\Delta x^\mu = -\sum_\nu \Gamma_{\mu\nu} A_\nu\,.
\tag{2.6}
$$

## 2.3   Drinfeld twists

As we expect that Abelian deformations correspond algebraically to Drinfeld twists, let us give their basic definition here: Consider the quasitriangular Hopf algebra $\mathcal{H}$ over the universal enveloping algebra $\mathcal{U}(\mathfrak{g})$ of a Lie algebra $\mathfrak{g}$. Denote the coproduct, counit, and antipode of $\mathcal{H}$ with $(\Delta, \varepsilon, \sigma)$, and its $R$ matrix by $R$. For details we refer to [39].

Now a *Drinfeld twist* $F = \sum_i f^i \otimes f_i$ is an invertible element of $\mathcal{U}(\mathfrak{g}) \otimes \mathcal{U}(\mathfrak{g})$ such that [10,40]

$$
\begin{aligned}
(F \otimes 1)(\Delta \otimes 1)F &= (1 \otimes F)(1 \otimes \Delta)F\,, \\
(\varepsilon \otimes 1)F &= (1 \otimes \varepsilon)F = 1 \otimes 1\,.
\end{aligned}
\tag{2.7}
$$

It can be used to define a modified (or twisted) quasitriangular Hopf algebra $\mathcal{H}_F$ with

$$
\begin{aligned}
\Delta_F(X) &= F\Delta(X)F^{-1}\,, \\
\sigma_F(X) &= u\sigma(X)u^{-1}\,, &&\text{with } u = \sum_i f^i \sigma(f_i)\,, \\
\varepsilon_F(X) &= \varepsilon(X)\,, \\
R_F(X) &= F_{21}RF^{-1}\,, &&\text{with } F_{21} = \sum_i f_i \otimes f^i\,.
\end{aligned}
\tag{2.8}
$$

The last relation will be of major interest to us, since the R matrix can typically be viewed as the scattering matrix of a physical integrable model. We expect the symmetry algebra of the homogeneous YB-deformed models to be Drinfeld twisted and therefore, generically, their S matrices to be related to the undeformed ones as the $R$ matrices are in equation (2.8). This expectation stems from the one-to-one relation between Drinfeld twists and solutions to the CYBE [10,11], see also [25]. In particular, for Abelian solutions $r$ the corresponding twist is

$$
F = \exp\left(\tfrac{i}{2h}r\right)\,.
\tag{2.9}
$$

We have introduced the inconsequential numerical factor $\frac{1}{2h}$ to ensure a direct match with the perturbative YB calculation. Note that in this special case $F_{21} = F^{-1}$.

One important subtlety here is that we only expect the full symmetry algebra $\mathfrak{psu}(2,2|4)$ to be Drinfeld twisted, while only $\mathfrak{su}(2|2)^{\oplus 2}_{\text{ce}}$ acts linearly on the S matrix of the gauge-fixed model. Hence we should distinguish the abstract formal $\mathfrak{psu}(2,2|4)$ $R$ matrix which we would expect to be Drinfeld twisted as described above, from the $\mathfrak{su}(2|2)^{\oplus 2}_{\text{ce}}$-invariant S matrix of the gauge-fixed model. For the latter it is not a priori obvious that, or how, it inherits this twist – clarifying this issue is the subject of this paper.

---

[3]See [12] for an explicit discussion from a Yang-Baxter model perspective.

## 2.4 Uniform light-cone gauge

For the quantization and calculation of the S matrix we fix a uniform light-cone gauge.[4] For this we pick two light-like isometry directions $x^\pm$ to form a light cone – in sections 3 and 4 we will explicitly state two different choices; these will lead to the BMN and GKP string respectively. For now we assume we made a choice and work out the general theory. In particular we allow for twisted boundary conditions of the light-cone directions, i.e. $\Delta x^\pm \neq 0$. The uniform light-cone gauge is then fixed by going to the first-order formalism and setting

$$x^+ = \tau + \Delta x^+ \frac{\sigma}{R_\sigma}, \qquad p_- = 1, \tag{2.10}$$

where $p_-$ is the conjugate momentum for $x^-$.

The extra $\Delta x^+$ term modifies the usual result for the gauge-fixed Lagrangian. As argued in appendix A (set $\xi = \frac{\Delta x^+}{R_\sigma}$) the gauge-fixed Lagrangian takes the form

$$\mathcal{L}_{\text{gf}}^{\Delta x^+} = \mathcal{L}_{\text{gf}}^0 /. \, \partial_\sigma x \to \partial_\sigma x - \frac{\Delta x^+}{R_\sigma} \partial_\tau x \quad \forall \text{ fields } x, \tag{2.11}$$

where $/.$ denotes replacement (as in Mathematica syntax) and $\mathcal{L}_{\text{gf}}^0$ is the usual gauge-fixed Lagrangian we get for $\Delta x^+ = 0$.

The two light-cone charges get affected by the gauge-fixing as follows: $P_+$ becomes related to the worldsheet Hamiltonian $H_{\text{ws}}$ through

$$P_+ = -H_{\text{ws}} \tag{2.12}$$

and with the gauge-fixing condition eq. (2.10) the generator $P_-$ becomes

$$P_- = \int_0^{R_\sigma} \mathrm{d}\sigma \, p_- = R_\sigma. \tag{2.13}$$

The Virasoro constraint $C_1$ is

$$\begin{aligned} C_1 = p_M x'^M &= p_+ x'^+ + p_- x'^- + p_a x'^a \\ &= p_+ \frac{1}{R_\sigma} \Delta x^+ + x'^- + p_a x'^a \overset{!}{=} 0, \end{aligned} \tag{2.14}$$

where the index $a$ runs over the directions transverse to the light cone. Integrating over $\sigma$ gives the level matching condition:

$$0 = \int \mathrm{d}\sigma \, C_1 = \frac{\Delta x^+}{R_\sigma} P_+ + \Delta x^- - P_{\text{ws}} \tag{2.15a}$$

$$\implies P_{\text{ws}} = \Delta x^- + \frac{\Delta x^+}{R_\sigma} P_+, \tag{2.15b}$$

where we used $\Delta x^- = \int \mathrm{d}\sigma \, x'^-$ and the total worldsheet momentum $P_{\text{ws}} = -\int \mathrm{d}\sigma \, p_a x'^a$.

**Alternative algorithm using T duality**    There is an alternative way of fixing the light-cone gauge. Instead of going to the first-order formalism and setting the momentum $p_- = 1$, we can also stay in the second-order formalism and use the isometry property of $x^-$ to T dualize in it. After that, in the T-dual theory, we set $\tilde{x}^-$ to be a winding mode with $\tilde{x}^- = \sigma$. This gives the exact same theory as the first-order procedure, but directly in the Lagrangian formalism.

---

[4] We closely follow the review [41].

## 2.5 Bethe equations

We want to compare the quantum theory of the YB-deformed models and the equivalent model with twisted boundary conditions. This is easily done at the level of the Bethe equations, as the boundary conditions can be implemented by an appropriate twist matrix here. We derive the Bethe equations by using the transfer matrix, following the asymptotic Bethe ansatz as presented in e.g. [42]. The transfer matrix is defined by[5]

$$t(p_A) = \mathrm{Tr}_A M_A \prod_j S_{Aj}(p_A, p_j), \tag{2.16}$$

where the subscript $A$ indicates the action on an auxiliary particle and $j$ runs over all particles on the Bethe chain. The twist matrix $M_A$ encodes possible twisted boundary conditions [26–28]. It is given by

$$M_A = \exp\left( i \sum_{\mu \neq \pm} \Delta x^\mu P_\mu^A \right), \tag{2.17}$$

where $\mu$ runs over all isometry directions that are twisted and $P_\mu^A$ only acts on the auxiliary particle.

The full Bethe equations follow from requiring that the eigenstates $|\psi\rangle$ of $t$ satisfy

$$e^{-ip_k R_\sigma} |\psi\rangle = -t(p_k)|\psi\rangle, \quad \forall k. \tag{2.18}$$

Observe that $t(p_k)$ becomes, using $S_{12}(p, p) = -\mathbb{P}_{12}$,

$$\begin{aligned}
t(p_k) &= \mathrm{Tr}_A M_A \prod_j S_{Aj}(p_k, p_j) \\
&= -\mathrm{Tr}_A M_A P_{Ak} \widehat{\prod_{j \neq k}} S_{kj}(p_k, p_j) \\
&= -M_k \widehat{\prod_{j \neq k}} S_{kj}(p_k, p_j).
\end{aligned} \tag{2.19}$$

Here the modified product is defined as

$$\widehat{\prod_{j \neq k}} S_{kj} = S_{kk+1} \dots S_{kN} S_{k1} \dots S_{kk-1}, \tag{2.20}$$

where $N$ is the total number of particles on the Bethe chain. With this, the Bethe equations become

$$e^{ip_k R_\sigma} M_k \widehat{\prod_{j \neq k}} S_{kj}(p_k, p_j)|\psi\rangle = |\psi\rangle, \quad \forall k. \tag{2.21}$$

In this paper we will not need the auxiliary Bethe equations describing the explicit diagonalization of the transfer matrix.

# 3 Deformed BMN string

We are now coming to the main section of the paper. We first discuss the perturbative calculation of the scattering matrix of the Abelian Yang-Baxter deformations. The result will split in two classes, depending on the presence of the light-cone generator $P_-$ in the r matrix. After that we

---

[5]We use the letter $p$ for both conjugate momenta and physical worldsheet momenta. We hope its meaning is clear from the context.

treat the deformation through the equivalent perspective of twisted boundary conditions. We find a match of both calculations at the level of their Bethe equations. This shows that, if $P_-$ is not present in the r matrix, the Drinfeld-twisted structure of Abelian deformations is preserved after gauge-fixing and quantization.

Let us start by stating our coordinate choices. For the BMN string we choose to parametrize $AdS_5 \times S^5$ through the coordinates and metric

$$
\begin{aligned}
ds^2 = &-(1+\rho^2)\,dt^2 + \frac{1}{1+\rho^2}\,d\rho^2 + \frac{\rho^2}{1-x^2}\,dx^2 + \rho^2(1-x^2)\,d\psi_1{}^2 + \rho^2 x^2\,d\psi_2{}^2 \\
&+ (1-r^2)\,d\phi^2 + \frac{1}{1-r^2}\,dr^2 + \frac{r^2}{1-w^2}\,dw^2 + r^2(1-w^2)\,d\phi_1{}^2 + r^2 w^2\,d\phi_2{}^2 \,.
\end{aligned}
\tag{3.1}
$$

We want to choose the light-cone directions to lie in $AdS_5$ and $S^5$. For the light-cone directions to lie in $AdS_5$ and $S^5$, we pick $t$ and $\phi$, with corresponding isometry generators $P_t$ and $P_\phi$, to form the light-cone coordinates

$$
\begin{aligned}
x^- &= \phi - t\,, & \phi &= x^+ + \tfrac{1}{2}x^-\,, \\
x^+ &= \tfrac{1}{2}(\phi + t)\,, & t &= x^+ - \tfrac{1}{2}x^-\,, \\[4pt]
P_- &= \tfrac{1}{2}(P_\phi - P_t)\,, & P_\phi &= \tfrac{1}{2}P_+ + P_-\,, \\
P_+ &= P_\phi + P_t\,, & P_t &= \tfrac{1}{2}P_+ - P_-\,.
\end{aligned}
\tag{3.2}
$$

We will exclusively use $x_\pm$ and $P_\pm$ from now on. When YB deforming the model we want to preserve the isometries needed for gauge-fixing. This is the case when the generators in the r matrix commute with $P_\pm$. The maximal subalgebra that does so is $\mathfrak{su}(2)^4 \oplus \mathbb{R} \oplus \mathbb{R}$. Its Cartan generators are the same as the ones of the full algebra and correspond to the six isometry directions explicit in the metric

$$
\{P_{\phi_1}, P_{\phi_2}, P_{\psi_1}, P_{\psi_2}, P_+, P_-\}\,,
\tag{3.3}
$$

see appendix B for their explicit definition. We are now going to look at Abelian deformations built from these Cartan generators.

## 3.1 Abelian Yang-Baxter deformations

We pick Cartan generators $P_\mu$, $P_\nu$ from the set (3.3) and combine them into a multiparameter r matrix

$$
r = \sum_{\mu\nu} \Gamma_{\mu\nu}\,\hat{P}_\mu \wedge \hat{P}_\nu\,,
\tag{3.4}
$$

where the antisymmetric matrix $\Gamma_{\mu\nu} = -\Gamma_{\nu\mu}$ contains the various deformation parameters. The hat on the generators indicates that they live in the Killing vector representation, see appendix B. The indices $\mu$, $\nu$ run over all six isometry directions. We use this r matrix to deform the Lagrangian as described in section 2.1 and gauge fix it as described in section 2.4. We then switch to coordinates $Y^{a\dot{a}}$ and $Z^{\alpha\dot{\alpha}}$ that are eigenstates of $\hat{P}_{\phi_{1,2}}$ and $\hat{P}_{\psi_{1,2}}$. Their definition is listed in eq. (B.2) in the appendix. Next we expand the gauge-fixed action to quartic order in fields, and lastly we use the result to compute the $2 \to 2$ tree-level T matrix, which is related to the full S matrix through an expansion in $\frac{1}{h}$

$$
S = 1 + \frac{i}{h}T + \dots \,.
\tag{3.5}
$$

Details of the calculation are laid out in appendix C and [9]. The results for the different parts of the r matrix split into two classes, depending on whether one of the $\hat{P}_{\mu,\nu}$ is $\hat{P}_-$ or not.

### 3.1.1 Deformations not including $P_-$

**One-parameter $\gamma$ deformation**   We start with the simple one-parameter case

$$r = \Gamma \hat{P}_{\phi_1} \wedge \hat{P}_{\phi_2}. \tag{3.6}$$

This corresponds to the one-parameter version of the $\gamma_i$ deformation whose scattering matrix has been presented in [29]. After following the steps as laid out before our calculation produces a deformed T matrix of the form

$$T^{\Gamma} = T^0 + \hat{T}, \tag{3.7}$$

where $T^0$ is the T matrix of the undeformed model [41,43][6] and the extra term is

$$\hat{T} = -\Gamma P_{\phi_1} \wedge P_{\phi_2}. \tag{3.8}$$

With the action of $P_{\phi_{1,2}}$ from eq. (B.7) this gives when acting on two-particle states

$$
\begin{aligned}
\hat{T}\left|Y_{1\dot{1}}Y_{1\dot{2}}\right\rangle, &= -\left|Y_{1\dot{1}}Y_{1\dot{2}}\right\rangle, & \hat{T}\left|Y_{1\dot{2}}Y_{1\dot{1}}\right\rangle, &= -\left|Y_{1\dot{2}}Y_{1\dot{1}}\right\rangle, \\
\hat{T}\left|Y_{1\dot{1}}Y_{2\dot{1}}\right\rangle, &= +\left|Y_{1\dot{1}}Y_{2\dot{1}}\right\rangle, & \hat{T}\left|Y_{2\dot{1}}Y_{1\dot{1}}\right\rangle, &= +\left|Y_{2\dot{1}}Y_{1\dot{1}}\right\rangle, \\
\hat{T}\left|Y_{2\dot{2}}Y_{1\dot{2}}\right\rangle, &= +\left|Y_{2\dot{2}}Y_{1\dot{2}}\right\rangle, & \hat{T}\left|Y_{1\dot{2}}Y_{2\dot{2}}\right\rangle, &= +\left|Y_{1\dot{2}}Y_{2\dot{2}}\right\rangle, \\
\hat{T}\left|Y_{2\dot{2}}Y_{2\dot{1}}\right\rangle, &= -\left|Y_{2\dot{2}}Y_{2\dot{1}}\right\rangle, & \hat{T}\left|Y_{2\dot{1}}Y_{2\dot{2}}\right\rangle, &= -\left|Y_{2\dot{1}}Y_{2\dot{2}}\right\rangle,
\end{aligned} \tag{3.9}
$$

and 0 otherwise. In [29] eq. (2.16) the four terms on the right-hand side where not presented explicitly, but they follow from the expression given in eq. (2.14) there. The full $T^{\Gamma}$ matrix matches the tree-level expansion of a Drinfeld-twisted full S matrix, as conjectured in [28] and described in section 2.3

$$S^{\Gamma} = F_{21}S^0 F^{-1}, \qquad \text{with } F = \exp\left(\frac{\mathrm{i}}{2h}\Gamma P_{\phi_1} \wedge P_{\phi_2}\right). \tag{3.10}$$

**General multi-term r matrix**   Let us generalize to an arbitrary and possibly multi-parameter deformation. We choose the r matrix

$$r = \sum_{\mu\nu\neq -} \Gamma_{\mu\nu} \hat{P}_{\mu} \wedge \hat{P}_{\nu}, \tag{3.11}$$

where $\Gamma_{\mu\nu}$ is an antisymmetric tensor of deformation parameters. The indices $\mu, \nu$ run over $\{\phi_1, \phi_2, \psi_1, \psi_2, +\}$. We discuss the case $\mu = -$ in the next section.

Now we can compute the perturbative T matrix following the steps laid out at the beginning of this subsection. Using the fact that combining Abelian Yang-Baxter deformations is a linear operation we disassembled the r matrix in its individual pieces and calculated the resulting T matrices for each term separately. Assembled back together we get the perturbative result

$$T^{\Gamma} = T^0 - \sum_{\mu\nu\neq -} \Gamma_{\mu\nu} P_{\mu} \wedge P_{\nu}, \tag{3.12}$$

with the generators acting as given in appendix B. The result is clearly not factorizable in the sense of eq. (2.111) of the review [41].

Again this matches the tree-level expansion of a Drinfeld-twisted full S matrix

$$S^{\Gamma} = F_{21}S^0 F^{-1}, \qquad \text{with } F = \exp\left(\frac{\mathrm{i}}{2h}\sum_{\mu\nu\neq -} \Gamma_{\mu\nu} P_{\mu} \wedge P_{\nu}\right). \tag{3.13}$$

---

[6]Even though the definition of $Y^{a\dot{a}}$ and $Z^{a\dot{a}}$ that the review [41] and we use differs from the definition in [43], the various bosonic T matrices all coincides.

It seems justified to assume that this is indeed the form of the full S matrix. However as mentioned we want to test this assumption by comparing to the equivalent model with twisted boundary conditions. We will do this comparison at the level of the Bethe equations. Hence we are now going to use this assumed form for the full YB-deformed S matrix to derive the corresponding Bethe equations.

**Bethe equations**   In section 2.5 the Bethe equations were laid out for the undeformed model. Starting again from the transfer matrix $t(p_A)$ and using $S^\Gamma$ for the S matrix we get

$$
\begin{aligned}
t(p_A) = \mathrm{Tr}_A \prod_j S^\Gamma_{Aj} &= \mathrm{Tr}_A \prod_j F_{jA} S^0_{Aj} F^{-1}_{Aj} \\
&\equiv \mathrm{Tr}_A \prod_j (F^{-1}_{Aj})^2 \prod_j S^0_{Aj} \, ,
\end{aligned}
\tag{3.14}
$$

where we used that under spectral equivalence $\equiv$ we have [28, 44]

$$
\begin{aligned}
S_{12} F_{13} &\equiv F_{13} S_{12} \, , \\
F_{12} F_{13} &\equiv F_{13} F_{12} \, .
\end{aligned}
\tag{3.15}
$$

We define and further simplify

$$
\begin{aligned}
N_A := \prod_j (F^{-1}_{Aj})^2 &= \prod_j \exp\left( -\mathrm{i} \sum_{\mu\nu\neq-} \Gamma_{\mu\nu} P^A_\mu P^j_\nu \right) \\
&= \exp\left( -\mathrm{i} \sum_{\mu\nu\neq-} \Gamma_{\mu\nu} P^A_\mu \sum_j P^j_\nu \right) \\
&= \exp\left( -\mathrm{i} \sum_{\mu\nu\neq-} \Gamma_{\mu\nu} P^A_\mu P_\nu \right) ,
\end{aligned}
\tag{3.16}
$$

where $P^j_\nu$ only acts on the $j$-th particle and $P_\nu$ on all particles. Again as argued in eq. (2.19) we get

$$
\begin{aligned}
t(p_k) = \mathrm{Tr}_A N_A \prod_j S_{Aj}(p_k, p_j) \\
= -\mathrm{Tr}_A N_A P_{Ak} \widehat{\prod_{j\neq k}} S_{kj}(p_k, p_j) \\
= -N_k \widehat{\prod_{j\neq k}} S_{kj}(p_k, p_j) ,
\end{aligned}
\tag{3.17}
$$

leading to the Bethe equations, cf. eq. (2.21)

$$
\mathrm{e}^{\mathrm{i}p_k R_\sigma} \exp\left( -\mathrm{i} \sum_{\mu\nu\neq-} \Gamma_{\mu\nu} P^k_\mu P_\nu \right) \widehat{\prod_{j\neq k}} S^0_{kj}(p_k, p_j) |\psi\rangle = |\psi\rangle \, , , \quad \forall k \, .
\tag{3.18}
$$

The level-matching condition takes the usual untwisted form

$$
P_{\mathrm{ws}} = \sum_j p_j = 0 \, .
\tag{3.19}
$$

We see that we get twisted Bethe equations from the twisted S matrix. In section 3.2 we will compare this to the Bethe equations derived from the model with undeformed Lagrangian but twisted boundary conditions on the fields. As shown there the final equations coincide.

**The case $P_{\mu,\nu} = P_+$** A comment is in order for the case that $r$ contains $P_+$. From the general calculations of the light-cone gauge, we know that $P_+ = -H_{\mathrm{ws}}$, see section 2.4. The perturbative calculations performed in the present section explicitly verify that this expectation also holds at the level of the Drinfeld twist. The generator acts on single particle states $|p\rangle = a^\dagger(p)|0\rangle$ of all particles types as

$$P_+ |p\rangle = -\omega_p |p\rangle \,, \tag{3.20}$$

where $\omega_p = \sqrt{1 + p^2}$ is the worldsheet energy. See appendix B for more details.

### 3.1.2 Deformations including $P_-$

Next we turn to all terms of the r matrix that contain $P_-$. Let us start with the one-parameter case and extend to the general case at the end. We pick the r matrix

$$r = \Gamma \hat{P}_- \wedge \hat{P}_\nu \,, \qquad \nu \in \{+, \phi_1, \phi_2, \psi_1, \psi_2\} \,, \tag{3.21}$$

and perform the steps mentioned at the beginning of this section. This leads to a deformed T matrix of the form

$$T^\Gamma(p_1, p_2) = T^0(p_1 - \Gamma P_\nu^1, p_2 - \Gamma P_\nu^2) \,. \tag{3.22}$$

We assume that we can extend this result to the full S matrix

$$S^\Gamma(p_1, p_2) = S^0(p_1 - \Gamma P_\nu^1, p_2 - \Gamma P_\nu^2) \,. \tag{3.23}$$

Again, we want to check this assumption and the Bethe equations following from it against the equivalent model with twisted boundary conditions. Before deriving the Bethe equations we want to derive the form of the full YB-deformed S matrix in eq. (3.23) in an alternative, more general way.

**General argument for a momentum shift** Assume that the coordinate $x^\nu$ corresponding to $P_\nu$ is an isometry direction (as it is for the present case). As a first step towards showing the claim (3.23) we observe that the deformed gauge-fixed Lagrangian can be expressed in terms of the undeformed one with a simple substitution

$$\mathcal{L}_{\mathrm{gf}}^\Gamma = \mathcal{L}_{\mathrm{gf}}^0 \, /. \, \partial_\sigma x \to \partial_\sigma x + \Gamma \hat{P}_\nu(x) \,, \quad \forall \text{ fields } x \,. \tag{3.24}$$

The argument goes as follows: As explained at the end of section 2.4, gauge fixing takes the form of a T-duality transform in $x^-$ followed by fixing $x^+ = \tau$ and $\tilde{x}^- = \sigma$. Let us denote these operations as $T_-, \iota_+, \iota_-$ respectively and write for the gauge-fixed Lagrangian

$$\mathcal{L}_{\mathrm{gf}} = \iota_+ \iota_- T_- \mathcal{L} \,. \tag{3.25}$$

Further, an Abelian YB deformations can be expressed as a TsT transformation, as explained in section 2.2. We express this transformation as

$$\mathcal{L}^\Gamma = T_- s^{\Gamma \tilde{x}^-} T_- \mathcal{L}^0 \,, \tag{3.26}$$

where $s^{\Gamma \tilde{x}^-}$ denotes the shift $x^\nu \to x^\nu + \Gamma \tilde{x}^-$. Now combining these two steps the gauge-fixed deformed Lagrangian is

$$\begin{aligned} \mathcal{L}_{\mathrm{gf}}^\Gamma = \iota_+ \iota_- T_- \mathcal{L}^\Gamma &= \iota_+ \iota_- T_- T_- s^{\Gamma \tilde{x}^-} T_- \mathcal{L}^0 \\ &= \iota_+ \iota_- s^{\Gamma \tilde{x}^-} T_- \mathcal{L}^0 \\ &= \iota_+ s^{\Gamma \sigma} \iota_- T_- \mathcal{L}^0 \,, \end{aligned} \tag{3.27}$$

where we used that T dualizing twice is a no-op, i.e. $T_- T_- = 1$. Further, in the last step we inserted $\tilde{x}^- = \sigma$ into the shift, which hence becomes

$$x^\nu \to x^\nu + \Gamma\sigma\,. \tag{3.28}$$

To finish the argument and identify the undeformed gauge-fixed Lagrangian $\mathcal{L}^0_{\text{gf}}$ we need to distinguish two cases:

$x^\nu = x^+$ We give a detailed explanation what happens when we set $x^+ = \tau + \Gamma\sigma$ in appendix A. The bottom line is that due to the general form of the T-dualized Lagrangian we are able to express the change happening due to the extra $\Gamma\sigma$ factor also as a replacement $\partial_\sigma \to \partial_\sigma - \Gamma\partial_\tau$ applied to the undeformed gauge-fixed Lagrangian. Stated as an equation this means

$$\mathcal{L}^\Gamma_{\text{gf}} = \mathcal{L}^0_{\text{gf}} / .\, \partial_\sigma x \to \partial_\sigma x - \Gamma\partial_\tau x\,, \quad \forall \text{ fields } x\,. \tag{3.29}$$

Now using $\hat{P}_+(x) = -\hat{H}_{\text{ws}}(x) = -\partial_\tau x$, see eq. (B.14), this is exactly our sought for statement (3.24):

$$\mathcal{L}^\Gamma_{\text{gf}} = \mathcal{L}^0_{\text{gf}} / .\, \partial_\sigma x \to \partial_\sigma x + \Gamma\hat{P}_+(x)\,, \quad \forall \text{ fields } x\,. \tag{3.30}$$

$x^\nu = \phi_{1,2}$ or $\psi_{1,2}$ In this case we can pull $\iota_+$ through the shift and identify $\mathcal{L}^0_{\text{gf}}$

$$\mathcal{L}^\Gamma_{\text{gf}} = s^{\Gamma\sigma}\iota_+\iota_- T_-\mathcal{L}^0 = s^{\Gamma\sigma}\mathcal{L}^0_{\text{gf}}\,. \tag{3.31}$$

Next we use that $x^\nu$ is an isometry direction, and hence only appears with derivatives acting on it. Effectively the shift (3.28) therefore becomes

$$\begin{aligned} \partial_\tau x^\nu &\to \partial_\tau x^\nu\,, \\ \partial_\sigma x^\nu &\to \partial_\sigma x^\nu + \Gamma\,. \end{aligned} \tag{3.32}$$

To find out how this shifts acts on arbitrary fields, even after a change of coordinates, e.g. to $Y^{a\dot{a}}$ and $Z^{\alpha\dot{\alpha}}$, we rewrite it using the translation operator $\hat{P}_\nu$:

$$\partial_\sigma x \to \partial_\sigma x + \Gamma\hat{P}_\nu(x)\,, \quad \forall \text{ fields } x\,. \tag{3.33}$$

In this form the statement indeed holds for all fields $x$ of the gauge-fixed Lagrangian in arbitrary coordinates and allows us to express eq. (3.31) as

$$\mathcal{L}^\Gamma_{\text{gf}} = \mathcal{L}^0_{\text{gf}} / .\, \partial_\sigma x \to \partial_\sigma x + \Gamma\hat{P}_\nu(x)\,, \quad \forall \text{ fields } x\,. \tag{3.34}$$

We arrive at the claimed statement that the deformed Lagrangian can be recovered from the undeformed one by a simple shift of $\partial_\sigma x$.

Now for the claimed form of the scattering matrix $S^\Gamma$, to which $\mathcal{L}^\Gamma_{\text{gf}}$ contributes in two ways. Firstly, the quadratic part gives the classical free solutions, i.e. the dispersion and asymptotic wave functions. The shifted form of the Lagrangian, here in momentum space ($x(\sigma) \to x(p)$),

$$\mathcal{L}^\Gamma_{\text{gf}} = \mathcal{L}^0_{\text{gf}} / .\, px(p) \to px(p) + i\Gamma\hat{P}_\nu(x(p)) \tag{3.35}$$

tells us that the deformed dispersion relation for classical solutions is

$$\omega^\Gamma(p) = \omega^0(p + i\Gamma\hat{P}_\nu(x)) \tag{3.36}$$

and similar for the wave function, which is however $p$ independent for the bosons in the present case. When transitioning to the quantum theory, the charges do not act by Poisson brackets

anymore, but rather directly as quantum operators. To get their action on asymptotic states we therefore have to multiply[7] by a factor of i, leading to the deformed dispersion relation for scattering states[8]

$$\omega^{\Gamma}(p) = \omega^0(p - \Gamma P_\nu). \tag{3.37}$$

The second contribution of $\mathcal{L}^{\Gamma}_{\mathrm{gf}}$ is through its remaining interaction terms, that give us $S^{\Gamma}$ through their Feynman rules. Due to the nature of these rules the simple shift of $\partial_\sigma$ or alternatively the momenta translates to

$$S^{\Gamma}(\{p_i, \omega^{\Gamma}_i\}) = S^0(\{p_i - \Gamma P^i_\nu, \omega^{\Gamma}_i\}), \tag{3.38}$$

where $\{p_i, \omega_i\}$ denotes the set of momenta and energies of all scattering particles. Again we multiplied the charges by a factor of i. Inserting eq. (3.37) gives

$$\begin{aligned}
S^{\Gamma}(\{p_i\}) = S^{\Gamma}(\{p_i, \omega^{\Gamma}_i(p_i)\}) &= S^0(\{p_i - \Gamma P^i_\nu, \omega^0_i(p_i - \Gamma P^i_\nu)\}) \\
&= S^0(\{p_i - \Gamma P^i_\nu\})
\end{aligned} \tag{3.39}$$

as claimed. Note that this analysis may miss subtleties in the quantum theory as it was solely happening at the level of the classical Lagrangian.[9]

Expressing $S^{\Gamma}$ in terms of $S^0$ comes with an interesting consequence: The undeformed scattering matrix for four particles can be expressed as

$$S^0(p_1, p_2, p_3, p_4) = S^0(p_1, p_2)\big(\delta(p_1 - p_3)\delta(p_2 - p_4) + \delta(p_1 - p_4)\delta(p_2 - p_3)\big), \tag{3.40}$$

reflecting the momentum conserving property of integrable scattering. Now with relation (3.39) this becomes for the deformed scattering matrix

$$\begin{aligned}
S^{\Gamma}(p_1, p_2, p_3, p_4) = {}& S^0(p_1 - \Gamma P^1_\nu, p_2 - \Gamma P^2_\nu) \\
&\times \Big(\delta(p_1 - \Gamma P^1_\nu - (p_3 - \Gamma P^3_\nu))\delta(p_2 - \Gamma P^2_\nu - (p_4 - \Gamma P^4_\nu)) \\
&+ \delta(p_1 - \Gamma P^1_\nu - (p_4 - \Gamma P^4_\nu))\delta(p_2 - \Gamma P^2_\nu - (p_3 - \Gamma P^3_\nu))\Big).
\end{aligned} \tag{3.41}$$

Hence we observe that not the momenta $p_i$ but their shifted versions $p_i - \Gamma P^i_\nu$ are conserved. The physical momenta $p_i$ itself jump proportionally to the charge difference.[10] For a discussion how this point derives in the perturbative calculation from evaluating the delta functions responsible for total momentum and energy conservation, see appendix C. The shift in the momenta also affects the Yang-Baxter equation, which is now satisfied in shifted form, as obviously inherited from the regular Yang-Baxter equation satisfied by the undeformed S matrix. Crossing and unitarity are compatible with the shift.

**Bethe equations**   The Bethe equations are[11]

$$e^{ip_k R_\sigma} \widehat{\prod_{j \neq k}} S^{\Gamma}_{kj}(p_k, p_j) |\psi\rangle = |\psi\rangle, \quad \forall k. \tag{3.42}$$

---

[7]The canonical quantization of $\hat{P}(\cdot) = \{\cdot, P\}$ leads to $-\mathrm{i}[\cdot, P]$. As we want $P$ to act directly on scattering states $|\psi\rangle$, we are only interested in the second term of the commutator multiplied by i.

[8]Directly calculating the dispersion relation from the quadratic terms of the YB-deformed Lagrangians gives the same result, as described in appendix C. For the case $\nu = +$ the relation for $\omega^{\Gamma}$ becomes recursive, as $P_+$ gives $\omega^{\Gamma}$, see eq. (B.14). The solution is given in eq. (C.6).

[9]We would like to thank Ben Hoare for pointing this out.

[10]Standard total momentum conservation in not affected by this due to conservation of the charge $P_\nu$.

[11]In the present case of a shifted momentum dependence of the S matrix we need to replace $t(p_k)$ by $t(p_k - \Gamma P^k_\nu)$ when deriving the Bethe equations from the transfer matrix in eq. (2.18).

Further expressing $S^\Gamma$ in terms of $S^0$ and introducing the shifted momenta $\bar{p}_k = p_k - \Gamma P_\nu^k$ gives

$$\mathrm{e}^{\mathrm{i}(\bar{p}_k + \Gamma P_\nu^k)R_\sigma} \widehat{\prod_{j \neq k}} S_{kj}^0(\bar{p}_k, \bar{p}_j) |\psi\rangle = |\psi\rangle \,, \quad \forall k \tag{3.43}$$

or alternatively, using $P_- = R_\sigma$,

$$\mathrm{e}^{\mathrm{i}\bar{p}_k R_\sigma} \exp\left(\mathrm{i}\Gamma P_\nu^k P_-\right) \widehat{\prod_{j \neq k}} S_{kj}^0(\bar{p}_k, \bar{p}_j) |\psi\rangle = |\psi\rangle \,, \quad \forall k \,. \tag{3.44}$$

The level matching turns into

$$P_{\mathrm{ws}} = \sum_j p_j = 0 \qquad \Longrightarrow \qquad \sum_j \bar{p}_j = -\Gamma P_\nu \,. \tag{3.45}$$

**General multi-term r matrix**  We can generalize this result to a linear combinations of deformations

$$r = \sum_\nu \Gamma_{-\nu} \hat{P}_- \wedge \hat{P}_\nu \,, \tag{3.46}$$

where $\nu \in \{\phi_1, \phi_2, \psi_1, \psi_2, +\}$. For such a deformation the scattering matrix is

$$S^\Gamma(p_1, p_2) = S^0\left(p_1 - \sum_\nu \Gamma_{-\nu} P_\nu^1, p_2 - \sum_\nu \Gamma_{-\nu} P_\nu^2\right). \tag{3.47}$$

The Bethe equation and level matching become in turn

$$\mathrm{e}^{\mathrm{i}\bar{p}_k R_\sigma} \exp\left(-\mathrm{i}\sum_\nu \Gamma_{\nu-} P_\nu^k P_-\right) \widehat{\prod_{j \neq k}} S_{kj}^0(\bar{p}_k, \bar{p}_j) |\psi\rangle = |\psi\rangle \,, \quad \forall k \,, \tag{3.48}$$

$$\sum_j \bar{p}_j = -\sum_\nu \Gamma_{-\nu} P_\nu \,, \tag{3.49}$$

where in the first line we used the antisymmetry of $\Gamma_{-\nu}$. This result will also be compared to the Bethe equations derived from the model with undeformed Lagrangian but twisted boundary conditions on the fields in section 3.2. Again the final equations coincide.

**Can the shift be expressed as a Drinfeld twist?**  We want to try to express the shifted S matrix through a standard Drinfeld twist. For this we rewrite

$$\begin{aligned}
S^\Gamma(p_1, p_2) &= S^0(p_1 - \Gamma P_\nu^1, p_2 - \Gamma P_\nu^2) \\
&= \mathrm{e}^{-\Gamma\left(P_\nu^1 \frac{\partial}{\partial p_1} + P_\nu^2 \frac{\partial}{\partial p_2}\right)} S^0(p_1, p_2) \mathrm{e}^{\Gamma\left(P_\nu^1 \frac{\partial}{\partial p_1} + P_\nu^2 \frac{\partial}{\partial p_2}\right)} \,.
\end{aligned} \tag{3.50}$$

Now compare this to the hypothetical Drinfeld-twisted S matrix

$$S^{\mathrm{Drinfeld}} = \mathrm{e}^{-\mathrm{i}\frac{\Gamma}{\hbar}\left(P_-^1 P_\nu^2 - P_\nu^1 P_-^2\right)} S^0(p_1, p_2) \mathrm{e}^{-\mathrm{i}\frac{\Gamma}{\hbar}\left(P_-^1 P_\nu^2 - P_\nu^1 P_-^2\right)} \,. \tag{3.51}$$

The important difference is that in the Drinfeld-twisted expression the generators of each term act on particle 1 and 2, while in the actual expression they both act on the same particle. Hence it appears impossible to express the shift as a Drinfeld twist.

## 3.2 Twisted boundary conditions

Performing an Abelian deformation on the model is classically equivalent to introducing twisted boundary conditions, see section 2.2. In this section we use this as an independent test of the results of the previous subsection. We will compute the Bethe equations for the model with undeformed Lagrangian but twisted boundary conditions and compare them to the Bethe equations for the deformed Lagrangian with untwisted boundary conditions obtained previously, finding perfect agreement.

The boundary conditions corresponding to the r matrix (3.4) are

$$\Delta x^\mu = -\sum_\nu \Gamma_{\mu\nu} P_\nu. \tag{3.52}$$

They imply the twist in the Bethe equations, as described in section 2.5,

$$M_A = \exp\left( i \sum_{\mu \neq \pm} \Delta x^\mu P_\mu^A \right). \tag{3.53}$$

Further we have to take into account that the presence of an $\Delta x^+$ requires us to modify the gauge fixing. As stated in eq. (2.10), we have to set $x^+ = \tau + \frac{\Delta x^+}{R_\sigma}\sigma$. This matches the situation described in appendix A with $\xi = \frac{\Delta x^+}{R_\sigma}$. This means that the gauge-fixed Lagrangian is of the shifted form

$$\mathcal{L}_{\text{gf}}^{\Delta x^+} = \mathcal{L}_{\text{gf}}^0 \, /. \, \partial_\sigma x \rightarrow \partial_\sigma x - \frac{\Delta x^+}{R_\sigma} \partial_\tau x \,. \tag{3.54}$$

The S matrix becomes by the same argument as in section 3.1.2

$$S^{\Delta x^+} = S^0(\{p_i - \frac{\Delta x^+}{R_\sigma} P_+^i\}) \tag{3.55}$$

and the level matching is, as derived in eq. (2.15b),

$$P_{\text{ws}} = \sum_j p_j \overset{!}{=} \Delta x^- + \frac{\Delta x^+}{R_\sigma} P_+ \,. \tag{3.56}$$

Now usually when calculating the S matrix in the decompactification limit we send $R_\sigma \to \infty$. However doing so in the present case and discarding the $\frac{1}{R_\sigma}$ terms in the S matrix and level matching would produce a mismatch with the YB case.[12] To remedy this mismatch we keep the $\frac{1}{R_\sigma}$ terms. As the asymptotic Bethe ansatz should account for all non-exponential $R_\sigma$ dependence, and further assumes that the scattering particles are well-separated, these terms become important – it is not necessary to take the strict decompactification limit in the level-matching and S matrix.

With this reasoning the Bethe equations become

$$e^{i p_k R_\sigma} \exp\left( i \sum_{\mu \neq \pm} \Delta x^\mu P_\mu^k \right) \widehat{\prod_{j \neq k}} S_{kj}^0(\{p_i - \frac{\Delta x^+}{R_\sigma} P_+^i\}) |\psi\rangle = |\psi\rangle \,, \tag{3.57}$$

which through introducing the shifted momenta

$$\bar{p}_k = p_k - \frac{\Delta x^+}{R_\sigma} P_+^k \tag{3.58}$$

---

[12]Explicitly we would miss the $e^{i \Delta x^+ P_+^k}$ term in the Bethe equations (3.59b). Interestingly, in it the $\frac{1}{R_\sigma}$ canceled with the $R_\sigma$ in $e^{i p_k R_\sigma}$.

becomes

$$e^{i\bar{p}_k R_\sigma + i\Delta x^+ P^k_+} \exp\left(i\sum_{\mu\neq\pm}\Delta x^\mu P^k_\mu\right)\widehat{\prod_{j\neq k}}S^0_{kj}(\bar{p}_k,\bar{p}_j)|\psi\rangle = |\psi\rangle \tag{3.59a}$$

$$\implies e^{i\bar{p}_k R_\sigma}\exp\left(i\sum_{\mu\neq-}\Delta x^\mu P^k_\mu\right)\widehat{\prod_{j\neq k}}S^0_{kj}(\bar{p}_k,\bar{p}_j)|\psi\rangle = |\psi\rangle \tag{3.59b}$$

$$\implies e^{i\bar{p}_k R_\sigma}\exp\left(-i\sum_{\substack{\mu\neq-\\\nu}}\Gamma_{\mu\nu}P^k_\mu P_\nu\right)\widehat{\prod_{j\neq k}}S^0_{kj}(\bar{p}_k,\bar{p}_j)|\psi\rangle = |\psi\rangle \,. \tag{3.59c}$$

The level-matching becomes

$$\sum_j p_j = \sum_j \bar{p}_j + \frac{\Delta x^+}{R_\sigma}\sum_j P^j_+ \overset{!}{=} \Delta x^- + \frac{\Delta x^+}{R_\sigma}P_+$$
$$\implies \sum_j \bar{p}_j = \Delta x^- = -\sum_\nu \Gamma_{-\nu}P_\nu \,. \tag{3.60}$$

Let us compare to the Yang-Baxter deformations.

**Comparison with YB deformations not including $P_-$**  To compare to the case where the r matrix does not contain $P_-$, we set $\Gamma_{-\nu} = \Gamma_{\nu-} = 0$. Then the Bethe equations (3.59c) and level-matching condition (3.60) become

$$e^{i\bar{p}_k R_\sigma}\exp\left(-i\sum_{\mu\nu\neq-}\Gamma_{\mu\nu}P^k_\mu P_\nu\right)\widehat{\prod_{j\neq k}}S^0_{kj}(\bar{p}_k,\bar{p}_j)|\psi\rangle = |\psi\rangle \,, \tag{3.61}$$

$$\sum_j \bar{p}_j = 0 \,. \tag{3.62}$$

This matches exactly the results for the YB deformation obtained in eqs. (3.18) and (3.19).

**Comparison with YB deformations including $P_-$**  Now for the case where the r matrix does contain $P_-$, we set $\Gamma_{-\nu} = -\Gamma_{\nu-}$ to be the only non-zero entries. This gives

$$e^{i\bar{p}_k R_\sigma}\exp\left(-i\sum_{\mu\neq-}\Gamma_{\mu-}P^k_\mu P_-\right)\widehat{\prod_{j\neq k}}S^0_{kj}(\bar{p}_k,\bar{p}_j)|\psi\rangle = |\psi\rangle \,, \tag{3.63}$$

$$\sum_j \bar{p}_j = -\sum_\nu \Gamma_{-\nu}P_\nu \,, \tag{3.64}$$

which again matches the results (3.48) for the YB deformation obtained earlier.

## 4  Deformed GKP string

The second part of the paper focuses on deformations that are most suitably analyzed in a different gauge. This results in a model that is related to the GKP string. Our efforts to calculate the S matrix for the deformations of this model we encountered unsolvable difficulties. Their nature will be apparent after we describe the S matrix calculation in the undeformed case and see how it generalizes to the deformed models.

However, let us start by giving the choice of light-cone directions we start with this time We want the them to lie purely in AdS$_5$, and hence start with the AdS$_5$ metric

$$ds^2 = \frac{1}{z^2}\left(\sum_{i=0}^{3}(dx^i)^2 + dz^2\right) \tag{4.1}$$

and ignore the S$^5$ part; it will not play any role in the following discussion. As light-cone directions we choose $x^0$ and $x^3$ with corresponding isometry generators $P_0$ and $P_3$. With this the light-cone coordinates and generators are

$$
\begin{aligned}
x^- &= x^3 - x^0, & x^3 &= x^+ + \tfrac{1}{2}x^-, \\
x^+ &= \tfrac{1}{2}(x^3 + x^0), & x^0 &= x^+ - \tfrac{1}{2}x^-, \\
P_- &= \tfrac{1}{2}(P_3 - P_0), & P_3 &= \tfrac{1}{2}P_+ + P_-, \\
P_+ &= P_3 + P_0, & P_0 &= \tfrac{1}{2}P_+ - P_-,
\end{aligned}
\tag{4.2}
$$

and again we will work with these instead of $x^0$ and $x^3$ from now on. Before we turn to the YB deformations in a second, we will review the S-matrix calculation in the undeformed case first.

## 4.1 Calculation of the undeformed S matrix

We present the undeformed calculation as it was done in [35, 36]. It will highlight why we will not be able to calculate an S matrix in the deformed case.

We start by taking the non-linear sigma model for the background eq. (4.1), Wick rotate, and gauge fix as described in section 2.4.[13] Then we introduce the complex coordinate

$$x = x^1 + ix^2 \qquad \bar{x} = x^1 - ix^2. \tag{4.3}$$

Now we observe that the gauge-fixed model has the following two solutions

- The ground state solution
$$z = \text{const}, \qquad x, \bar{x} = 0, \tag{4.4}$$
which however gives a massless theory. Due to complications arising in the calculation and interpretation of massless S matrices in two dimensions [45] this classical solution is not widely used as a starting point for quantization in the literature.

- The null-cusp solution
$$z = \sqrt{\frac{\tau}{\sigma}}, \qquad x, \bar{x} = 0, \tag{4.5}$$
which gives a massive theory, and is related to the GKP string. We will use this solution as the starting point for the calculation of an S matrix.

The classical solution (4.5) depends on the worldsheet coordinates. Therefore, expanding around it leads to an inhomogeneous Lagrangian, i.e. one that also depends on the worldsheet coordinates. This makes the momentum space methods in the usual perturbative calculation of the S matrix useless. To circumvent this problem we redefine the string coordinates to

$$z \to \sqrt{\frac{\tau}{\sigma}}z, \qquad x \to \sqrt{\frac{\tau}{\sigma}}x, \tag{4.6}$$

---

[13]The authors of [35] choose the alternative gauge $\sqrt{-g}g^{\alpha\beta} = \text{diag}(-z^2, z^{-2})$. Our choice results in the same action in the undeformed case.

and the worldsheet coordinates to

$$t = \ln \tau \, , \qquad s = \ln \sigma \, , \qquad \mathrm{d}t \, \mathrm{d}s = \frac{\mathrm{d}\tau \, \mathrm{d}\sigma}{\tau \sigma} \, , \qquad \tau \partial_\tau = \partial_t \, , \qquad \sigma \partial_\sigma = \partial_s \, . \qquad (4.7)$$

Now we have the Lagrangian

$$\mathcal{L} = \left| \partial_t x + \tfrac{1}{2} x \right|^2 + \frac{1}{z^4} \left| \partial_s x - \tfrac{1}{2} x \right|^2 + \left( \partial_t z + \tfrac{1}{2} z \right)^2 + \frac{1}{z^4} \left( \partial_s z - \tfrac{1}{2} z \right)^2 \, , \qquad (4.8)$$

that we can expand around the null-cusp solution (4.5), which now reads

$$z = 1 \, , \qquad x = 0 \, , \qquad (4.9)$$

and calculate the S matrix as was done in [36].

The crucial point in the success of this calculation is that the rescaling eq. (4.6) removes the worldsheet dependence of the Lagrangian introduced by the null-cusp solution (4.5). As we will see, this is no longer given for the various deformed cases we consider next.

## 4.2 Abelian Yang-Baxter deformations

For the Yang-Baxter deformation we pick two generators $P_\mu$, $P_\nu$ from the set

$$\{ P_+, P_-, P_x, P_{\bar{x}}, P_{12} \} \, , \qquad (4.10)$$

where $P_\pm$ generate shifts in $x^\pm$, $P_x$ generates shifts in $x$, $P_{\bar{x}}$ generates shifts in $\bar{x}$, and $P_{12}$ rotates the $x^1$-$x^2$ plane. The coordinates $x, \bar{x}$ are thus eigendirections of $P_{12}$. We combine two of these generators into the r matrix

$$r = \Gamma P_\mu \wedge P_\nu \qquad (4.11)$$

with deformation parameter $\Gamma$. We use the r matrix to deform the Lagrangian as described in section 2.1, and then proceed as in the previous section 4.1: We gauge fix it as described in section 2.4 and pick the classical solution derived from the null-cusp solution eq. (4.5). But then we fail to find a rescaling à la eqs. (4.6) and (4.7) that renders the Lagrangian independent of the worldsheet coordinates. In the following we discuss why for a few examples of different $P_\mu \wedge P_\nu$.

**$P_x \wedge P_{\bar{x}}$** The deformed metric and B field are given by

$$\begin{aligned}
\mathrm{d}s^2 &= \frac{1}{z^2} \, \mathrm{d}x^+ \mathrm{d}x^- + \frac{z^2}{z^4 + \Gamma^2} \, \mathrm{d}x \, \mathrm{d}\bar{x} + \frac{1}{z^2} \, \mathrm{d}z^2 \, , \\
B &= \frac{\mathrm{i}\Gamma}{z^4 + \Gamma^2} \, \mathrm{d}x \wedge \mathrm{d}\bar{x} \, .
\end{aligned} \qquad (4.12)$$

As in the undeformed case we gauge fix and Wick rotate arriving at the Lagrangian

$$\begin{aligned}
\mathcal{L} = \frac{1}{2} &\left( \frac{z^4}{z^4 + \Gamma^2} |\partial_\tau x|^2 + \frac{1}{z^4 + \Gamma^2} |\partial_\sigma x|^2 + (\partial_\tau z)^2 + \frac{1}{z^4} (\partial_\sigma z)^2 \right) \\
&- \frac{1}{2} \frac{\Gamma}{z^4 + \Gamma^2} (\partial_\tau x \partial_\sigma \bar{x} - \partial_\sigma x \partial_\tau \bar{x}) \, .
\end{aligned} \qquad (4.13)$$

The null-cusp solution (4.5) is still a valid solution. If we now rescale the fields $x$, $\bar{x}$ and $z_i$ by the null-cusp expression $\sqrt{\frac{\tau}{\sigma}}$ and change to the worldsheet coordinates $s$ and $t$ as in eq. (4.7),

we get the Lagrangian

$$\mathcal{L} = \frac{1}{2}\left(\frac{z^4}{z^4+\Gamma^2/\frac{\tau^2}{\sigma^2}}\left|\partial_t x + \tfrac{1}{2}x\right|^2 + \frac{1}{z^4+\Gamma^2/\frac{\tau^2}{\sigma^2}}\left|\partial_s x - \tfrac{1}{2}x\right|^2 + (\partial_\tau z + \tfrac{1}{2}z)^2 + \frac{1}{z^4}(\partial_s z - \tfrac{1}{2}z)^2\right)$$
$$-\frac{1}{2}\frac{\Gamma/\frac{\tau}{\sigma}}{z^4+\Gamma^2/\frac{\tau^2}{\sigma^2}}\left((\partial_t x + \tfrac{1}{2}x)(\partial_s \bar{x} - \tfrac{1}{2}\bar{x}) - (\partial_s x - \tfrac{1}{2}x)(\partial_t \bar{x} + \tfrac{1}{2}\bar{x})\right).$$
(4.14)

We see that the Lagrangian is not homogeneous, i.e. it has worldsheet-dependent terms, and therefore does not allow a direct computation of the S matrix. We were also not able to find another field redefinition that removes these terms.

**$P_- \wedge P_{12}$**    Analogously to the BMN case, we get a deformed Lagrangian that is a shifted version of the undeformed one

$$\mathcal{L}^\Gamma = \mathcal{L}^0 /. \{\partial_\sigma x \to \partial_\sigma x + \Gamma x, \partial_\sigma \bar{x} \to \partial_\sigma \bar{x} - \Gamma \bar{x}\}$$
(4.15)

due to the fields $x$ and $\bar{x}$ being eigenvectors of $P_{12}$. The null-cusp solution (4.5) still gives a valid solution. After rescaling by $\sqrt{\frac{\tau}{\sigma}}$ and changing to the worldsheet coordinates $s$ and $t$ the gauge-fixed Lagrangian becomes

$$\mathcal{L}^\Gamma = \left|\partial_t x + \tfrac{1}{2}x\right|^2 + \frac{1}{z^4}\left|\partial_s x - \tfrac{1}{2}x \pm \Gamma e^s x\right|^2 + \left(\partial_t x + \tfrac{1}{2}x\right)^2 + \frac{1}{z^4}\left(\partial_s x - \tfrac{1}{2}x\right)^2.$$
(4.16)

We see that the Lagrangian is not homogeneous, i.e. it has worldsheet-dependent terms, and therefore does not allow a direct computation of the S matrix. Note that due to the coordinate rescaling we cannot apply the reasoning of section 3.1.2 to derive an S matrix with shifted momenta from the replacement (4.15). We were also not able to find another field redefinition that removes these terms. The only field redefinition we found to be useful is

$$x \to \exp(-\Gamma e^s)x, \qquad \bar{x} \to \exp(\Gamma e^s)\bar{x}.$$
(4.17)

However this completely removes the deformation from the Lagrangian and shifts it into the boundary conditions of the model, giving back the undeformed model with twisted boundary conditions that we expect from the discussion at the end of section 2.2. In the original $\sigma$ coordinates we would be able to express the deformation through a shift of the $\sigma$-momenta. However in these coordinates we do not know the S matrix. In the new $s$ coordinates we do not know how to handle the deformation sensibly, as it does not give a simple shift, even though the S matrix is known.

**$P_- \wedge P_+$**    Again the deformed gauge-fixed Lagrangian can be retrieved from the undeformed one, this time with the replacement[14]

$$\mathcal{L}^\Gamma = \mathcal{L}^0 /. \partial_\sigma \to \partial_\sigma - i\Gamma \partial_\tau.$$
(4.18)

After this shift the deformed Lagrangian has the modified null-cusp solution

$$\sqrt{\frac{\tau + i\Gamma\sigma}{\sigma}}.$$
(4.19)

---

[14]The factor of i is due to the Wick rotation.

Rescaling by this expression instead of $\sqrt{\frac{\tau}{\sigma}}$ and changing to the worldsheet coordinates $s$ and $t$, the gauge-fixed Lagrangian becomes

$$
\begin{aligned}
\mathcal{L}^{\Gamma} = \frac{1}{2} \frac{1}{1+\mathrm{i}\Gamma e^{s-t}} \Big( & \big((1+\mathrm{i}\Gamma e^{s-t})\partial_t x + \tfrac{1}{2}x\big)\big((1+\mathrm{i}\Gamma e^{s-t})\partial_t \bar{x} + \tfrac{1}{2}\bar{x}\big) \\
& + \frac{1}{z^4}\big(\partial_s x - \tfrac{1}{2}x - \mathrm{i}\Gamma e^{s-t}\partial_t x\big)\big(\partial_s \bar{x} - \tfrac{1}{2}\bar{x} - \mathrm{i}\Gamma e^{s-t}\partial_t \bar{x}\big) \\
& + \big((1+\mathrm{i}\Gamma e^{s-t})\partial_t z + \tfrac{1}{2}z\big)^2 + \frac{1}{z^4}\big(\partial_s z - \tfrac{1}{2}z - \mathrm{i}\Gamma e^{s-t}\partial_t z\big)^2 \Big).
\end{aligned}
\tag{4.20}
$$

Again the Lagrangian is not homogeneous, i.e. it has worldsheet-dependent terms, and therefore does not allow a direct computation of the S matrix. We were also not able to find another field redefinition that removes these terms.

**Summary**   It seems that the undeformed Lagrangian is of a particular fine-tuned form that subsequently allows the redefinitions eqs. (4.6) and (4.7) to remove any worldsheet dependence introduced through expanding around the null-cusp solution (4.5). Any YB deformation destroys this fine tuning: Expanding around the (modified) null-cusp solution inevitably introduces a worldsheet dependence that we were not able to remove by further coordinate redefinitions. This worldsheet dependence then renders the momentum space methods of the usual perturbative calculations useless and we are not able to compute a scattering matrix. Of course we cannot exclude that it is possible to find a coordinate redefinition that circumvents these problems.

# 5   Conclusion

We approached the question posed in the introduction *How does gauge fixing affect the expected Drinfeld twist of the S matrix?* from two different angles. Firstly, we directly calculated the tree-level T matrix from the deformed Lagrangian. Secondly, we determined the Bethe equations for the equivalent undeformed model with twisted boundary conditions. For the BMN string we find the following S matrices that are in accordance with both approaches:

**r matrix not including $P_-$**   We find an S matrix with Drinfeld twist as in eq. (3.13)

$$
S^{\Gamma} = F_{21} S^0 F^{-1}, \qquad \text{with } F = \exp\Big(\frac{\mathrm{i}}{2h}\sum_{\mu\nu}\Gamma_{\mu\nu}P_\mu \wedge P_\nu\Big),
\tag{5.1}
$$

for r matrices (3.11) that contain the non-light-cone shift generators $P_{\phi_{1,2}}, P_{\psi_{1,2}}$ or the light-cone shift generator $P_+$. The latter acts as the worldsheet Hamiltonian, $P_+ = -H_{\mathrm{ws}}$, on the spectrum of the gauge-fixed string.

**r matrix including $P_-$**   We find an S matrix with momentum dependence shifted

$$
S^{\Gamma}(p_1, p_2) = S^0\big(p_1 - \sum_\nu \Gamma_{-\nu}P_\nu^1, p_2 - \sum_\nu \Gamma_{-\nu}P_\nu^2\big),
\tag{5.2}
$$

see eq. (3.47), for r matrices (3.46) that contain the light-cone generator $P_-$.

The first result extends what is known about the one-parameter $\gamma_i$ deformation to more Abelian Yang-Baxter deformations. In particular it matches perfectly with the Drinfeld-twisted structure found previously for homogeneous Yang-Baxter deformations. The momentum shift induced by deformations involving the light-cone $x^-$ direction has not been observed before to our knowledge. Interestingly, the resulting S matrix satisfies not the usual, but a shifted

version of the (quantum) Yang-Baxter equation.[15] Of course, our notion of a deformation involving $x^-$ is dependent on the choice of light-cone gauge (coordinates), and hence for e.g. single-parameter $\gamma_i$ deformations, it is always possible to fix a light-cone gauge such that the deformation does not involve $x^-$. In other words, the momentum shift of the $S$ matrix must be equivalent to a proper Drinfeld twist at the level of the full energy spectrum at least.[16] This spectral equivalence is not manifest in the Bethe ansatz built on the $\mathfrak{su}(2|2)_{\text{ce}}^{\oplus 2}$ S matrix, but in the quantum spectral curve for e.g. the $\gamma_i$ deformation, the deformation parameters and charges enter in a "symmetric" fashion [46]. Of course, for general multi-parameter deformation it is impossible to avoid the momentum shift in the S matrix.

The second part of the paper is concerned with the GKP string, see section 4. We tried to calculate the S matrix and Bethe equations as well, however have to report a negative result. Our attempts failed because of a worldsheet-dependent field redefinition $x \to \sqrt{\frac{\tau}{\sigma}}x$ that is necessary already in the undeformed case but does not generalize to the deformed one. More precisely, while the redefinition is not able to remove all worldsheet dependence from the Lagrangian – making perturbative calculations impossible – it also causes the symmetry charges to become worldsheet-dependent themselves. This prevents us from writing down sensible twisted Bethe equations. So neither the direct perturbative approach, nor the twisted-boundary-condition approach can be straightforwardly applied.

In future work if would be interesting to investigate the following issues. For Abelian deformations of the BMN string we treated all possible r matrices consisting of two bosonic generators that preserve the light-cone isometries needed for gauge-fixing. This leaves us with r matrices that contain fermionic generators. It would be interesting to study such fermionic Abelian deformations to clarify if they preserve the light-cone isometries and what their physical interpretation is. Going beyond the BMN case, we would like to resolve the issues of the deformed GKP string and find a way to derive its scattering matrix. Another problem of great interest is the treatment of Abelian (and general YB) deformations which break the light-cone isometries and hence make the usual light-cone gauge fixing impossible. Finding a way to nevertheless be able to compute the scattering matrix of these models, or an alternative quantization approach, would be invaluable for a wide range of applications. Moreover, more general deformed sigma models are still actively being uncovered, see e.g. [6, 47, 48], and it would be insightful to investigate them from an integrable S matrix perspective where possible. Lastly a connection to the field theory side, in particular a calculation of the spectrum in (Cartan-twisted) noncommutative SYM and subsequent matching would be highly desirable for extending the known AdS/CFT dictionary.

# Acknowledgments

We would like to thank Ben Hoare and Alessandro Sfondrini for discussions, and Changrim Ahn, Riccardo Borsato, Ben Hoare, and Minkyoo Kim for comments on the draft of this paper. The work of ST and YZ is supported by the German Research Foundation (DFG) via the Emmy Noether program "Exact Results in Extended Holography". YZ would like to thank his parents

---

[15]It seems reasonable to assume that quantum integrability in the form of infinitely many conserved charges is inherited from the undeformed theory under the momentum shift (3.35), and hence still present. In this picture the charges gain a particle species dependence through the operator-valued shift of momenta, however, which presumably leads to the shifted version of the Yang-Baxter equation.

[16]In a related context, the S matrices for various inequivalent inhomogeneous deformations of $AdS_5 \times S^5$ are manifestly spectrally equivalent, being related by one-particle momentum-dependent changes of basis, at least at tree level [31]. In the present context, given the complicated functional form of the S matrix, it seems unlikely that a momentum shift can be traded for a Drinfeld twist and a possibly momentum dependent basis change, but we have not investigated this question in detail.

for their warm-hearted hospitality during the finalization of the manuscript and Lena Konst for her technical assistance. ST is supported by LT.

## A  The effect of fixing $x^+ = \tau + \xi\sigma$

In the light-cone gauge we usually set $x^+ = \tau$. In this appendix we want to discuss how setting $x^+ = \tau + \xi\sigma$, with $\xi$ a free parameter, affects the gauge-fixed Lagrangian and the S matrix. This situation occurs for the $r = P_- \wedge P_+$ YB deformation of section 3.1.2 and for the model with twisted boundary conditions of section 3.2.

**Effect on the Lagrangian**  Recall that light-cone gauge fixing can be done in the first-order Hamiltonian formalism or alternatively in the second-order Lagrangian formalism, by T dualizing in $x^-$ and fixing $x^+ = \tau$ and $\tilde{x}^- = \sigma$ in the resulting Lagrangian. Let us now look at exactly this T dualized Lagrangian before fixing $x^+$ and $\tilde{x}^-$. After fixing the worldsheet metric it takes the form [49]

$$\mathcal{L}_{\text{T-dual}} = -2\sqrt{G} - E, \tag{A.1}$$

where

$$
\begin{aligned}
G &= \det(G_{\alpha\beta}), & E &= \epsilon^{\alpha\beta} E_{\alpha\beta}, \\
G_{\alpha\beta} &= \mathring{g}_{MN} \partial_\alpha x^M \partial_\beta x^N, & E_{\alpha\beta} &= \mathring{B}_{MN} \partial_\alpha x^M \partial_\beta x^N,
\end{aligned}
\tag{A.2}
$$

and $\mathring{g}_{MN}$ and $\mathring{B}_{MN}$ are the T-dual metric and B field. The indices $M, N \in \{+, -, a\}$ where $a$ designates the transverse coordinates; and the indices $\alpha, \beta \in \{\tau, \sigma\}$. We want to show now that the effect of setting $x^+ = \tau + \xi\sigma$ can equally be achieved by keeping $x^+ = \tau$ but replacing $\partial_\sigma x^a \to \partial_\sigma x^a - \xi \partial_\tau x^a$. We rewrite[17]

$$
\begin{aligned}
E &= E_{\tau\sigma} - E_{\sigma\tau} = \mathring{B}_{MN} \partial_\tau x^{[M} \partial_\sigma x^{N]}, \\
G &= G_{\tau\tau} G_{\sigma\sigma} - G_{\tau\sigma} G_{\sigma\tau} \\
&= \mathring{g}_{MN} \mathring{g}_{OP} \big( \partial_\tau x^M \partial_\tau x^N \partial_\sigma x^O \partial_\sigma x^P - \partial_\tau x^M \partial_\sigma x^N \partial_\sigma x^O \partial_\tau x^P \big) \\
&= \mathring{g}_{MN} \mathring{g}_{OP} \partial_\tau x^M \partial_\sigma x^O \partial_\tau x^{[N} \partial_\sigma x^{P]} \\
&= \frac{1}{2} \mathring{g}_{MN} \mathring{g}_{OP} \partial_\tau x^{[M} \partial_\sigma x^{O]} \partial_\tau x^{[N} \partial_\sigma x^{P]},
\end{aligned}
\tag{A.3}
$$

where in the last step we used the symmetry of $\mathring{g}_{MN}$. We see that the derivative terms come only in combinations $\partial_\tau x^{[M} \partial_\sigma x^{N]}$. Further $\mathring{g}_{MN}$ is independent of $x^+$ because we require that $x^+$ is an isometry direction. So to compare the effect of setting $x^+ = \tau + \xi\sigma$ to the replacement of derivatives we just need to check the different $\partial_\tau x^{[M} \partial_\sigma x^{N]}$ terms.

- For $\partial_\tau x^{[+} \partial_\sigma x^{a]} = \partial_\tau x^+ \partial_\sigma x^a - \partial_\tau x^a \partial_\sigma x^+$ setting $x^+ = \tau + \xi\sigma$ gives the same result as keeping $x^+ = \tau$ and replacing $\partial_\sigma x^a \to \partial_\sigma x^a - \xi \partial_\tau x^a$.

- For $\partial_\tau x^{[a} \partial_\sigma x^{v]} = \partial_\tau x^a \partial_\sigma x^v - \partial_\tau x^v \partial_\sigma x^a$ nothing changes in both cases; any extra terms cancel.

- For $\partial_\tau x^{[-} \partial_\sigma x^{M]} = \partial_\tau x^- \partial_\sigma x^M - \partial_\tau x^M \partial_\sigma x^-$ the first term becomes zero upon setting $x^- = \sigma$. Therefore the $\partial_\sigma x^M$ term does not contribute at all.

We see that for $G$ and $E$ fixing $x^+$ differently has the same effect as the mentioned replacement, so we write for the full Lagrangian (A.1) after inserting $x^+ = \tau$ and $\tilde{x}^- = \sigma$

$$\mathcal{L}_{\text{gf}}^\xi = \mathcal{L}_{\text{gf}}^0 \,/.\, \partial_\sigma x \to \partial_\sigma x - \xi \partial_\tau x, \quad \forall \text{ fields } x, \tag{A.4}$$

---

[17]Define $x^{[MN]} = x^{MN} - x^{NM}$.

where $\mathcal{L}^0_{\text{gf}}$ is the usual gauge-fixed Lagrangian we get for $x^+ = \tau$, i.e. $\xi = 0$. The Lagrangian only contains the transverse fields.

**Effect on the S matrix**   We can write the relation from eq. (A.4) in momentum space by replacing every occurrence of the worldsheet momenta[18]

$$\mathcal{L}^\xi = \mathcal{L}^0 \, / . \, p\tilde{x} \to p\tilde{x} + \xi\omega\tilde{x} \, , \tag{A.5}$$

which then gives by the same argument as presented in section 3.1.2 the S matrix

$$S^\xi(\{p_i\}) = S^0(\{p_i + \xi\omega_i\}) = S^0(\{p_i - \xi P^i_+\}) \, . \tag{A.6}$$

## B   Action of *P*'s

The Cartan generators $\{P_{\phi_1}, P_{\phi_2}, P_{\psi_1}, P_{\psi_2}, P_+, P_-\}$ of the symmetry algebra of the BMN string split in two groups: two generators $P_\pm$ associated with the light-cone directions and four generators $P_{\psi_{1,2}}, P_{\phi_{1,2}}$ not associated with them. We give their explicit action in turn. The latter correspond to shift isometries in the variables $\psi_{1,2}, \phi_{1,2}$ respectively. Their actions on the coordinate fields is therefore given by

$$\begin{aligned}
\hat{P}_{\phi_1}(\phi_1) &= 1 \, , & \hat{P}_{\phi_2}(\phi_2) &= 1 \, , \\
\hat{P}_{\psi_1}(\psi_1) &= 1 \, , & \hat{P}_{\psi_2}(\psi_2) &= 1 \, ,
\end{aligned} \tag{B.1}$$

and 0 otherwise. Next we introduce the coordinates $Y^{a\dot{a}}, Z^{\alpha\dot{\alpha}}$ which are eigenstates of these operators. They are defined as[19]

$$\begin{aligned}
Y^{1\dot{1}} &= \frac{1}{r}\Big(1 - \sqrt{1 - r^2}\Big)\sqrt{1 - w^2}e^{-i\phi_1} \, , & Y^{1\dot{2}} &= \frac{1}{r}\Big(1 - \sqrt{1 - r^2}\Big)we^{-i\phi_2} \, , \\
Y^{2\dot{2}} &= \frac{1}{r}\Big(1 - \sqrt{1 - r^2}\Big)\sqrt{1 - w^2}e^{+i\phi_1} \, , & Y^{2\dot{1}} &= \frac{1}{r}\Big(1 - \sqrt{1 - r^2}\Big)we^{+i\phi_2} \, , \\
Z^{3\dot{3}} &= \frac{1}{\rho}\Big(-1 + \sqrt{1 + \rho^2}\Big)\sqrt{1 - x^2}e^{-i\psi_1} \, , & Z^{3\dot{4}} &= \frac{1}{\rho}\Big(-1 + \sqrt{1 + \rho^2}\Big)xe^{-i\psi_2} \, , \\
Z^{4\dot{4}} &= \frac{1}{\rho}\Big(-1 + \sqrt{1 + \rho^2}\Big)\sqrt{1 - x^2}e^{+i\psi_1} \, , & Z^{4\dot{3}} &= \frac{1}{\rho}\Big(-1 + \sqrt{1 + \rho^2}\Big)xe^{+i\psi_2} \, .
\end{aligned} \tag{B.2}$$

The action of the shift operators becomes

$$\begin{aligned}
\hat{P}_{\phi_1}(Y^{1\dot{1}}) &= -iY^{1\dot{1}} \, , & \hat{P}_{\phi_2}(Y^{1\dot{2}}) &= -iY^{1\dot{2}} \, , \\
\hat{P}_{\phi_1}(Y^{2\dot{2}}) &= +iY^{2\dot{2}} \, , & \hat{P}_{\phi_2}(Y^{2\dot{1}}) &= +iY^{2\dot{1}} \, , \\[6pt]
\hat{P}_{\psi_1}(Z^{3\dot{3}}) &= -iZ^{3\dot{3}} \, , & \hat{P}_{\psi_2}(Z^{3\dot{4}}) &= -iZ^{3\dot{4}} \, , \\
\hat{P}_{\psi_1}(Z^{4\dot{4}}) &= +iZ^{4\dot{4}} \, , & \hat{P}_{\psi_2}(Z^{4\dot{3}}) &= +iZ^{4\dot{3}} \, .
\end{aligned} \tag{B.3}$$

The corresponding Noether charges for the gauge-fixed model are

$$P_\mu = \int d\sigma \, p_M \hat{P}_\mu(x^M) \, , \tag{B.4}$$

---

[18]Note that $\partial_\tau \to i\omega$ and $\partial_\sigma \to -ip$.

[19]This corresponds to a choice of $Y^{a\dot{a}}$ and $Z^{\alpha\dot{\alpha}}$ as in the review [41].

with $p_M = \frac{\partial \mathcal{L}_{\text{gf,2}}}{\partial(\partial_\tau x^M)}$ the conjugate momenta for the gauge-fixed, free Lagrangian. The charges act through the Poisson bracket, giving back the original generators

$$\hat{P}_\mu(\cdot) = \{\cdot, P_\mu\}.\tag{B.5}$$

As we turn to the quantum theory, we use the mode expansions of appendix C to express the charges through annihilation and creation operators:

$$
\begin{aligned}
P_{\phi_1} &= \int \mathrm{d}p \left(a^\dagger_{1\dot{1}} a^{1\dot{1}} - a^\dagger_{2\dot{2}} a^{2\dot{2}}\right),\\
P_{\phi_2} &= \int \mathrm{d}p \left(a^\dagger_{1\dot{2}} a^{1\dot{2}} - a^\dagger_{2\dot{1}} a^{2\dot{1}}\right),\\
P_{\psi_1} &= \int \mathrm{d}p \left(a^\dagger_{3\dot{3}} a^{3\dot{3}} - a^\dagger_{4\dot{4}} a^{4\dot{4}}\right),\\
P_{\psi_2} &= \int \mathrm{d}p \left(a^\dagger_{3\dot{4}} a^{3\dot{4}} - a^\dagger_{4\dot{3}} a^{4\dot{3}}\right),
\end{aligned}
\tag{B.6}
$$

where the $p$ dependence of the operators is hidden. Using $[a^{M\dot{M}}(p), a^\dagger_{N\dot{N}}(q)] = \delta^M_N \delta^{\dot{M}}_{\dot{N}} \delta(p-q)$ with $M\dot{M}$, $N\dot{N}$ running over the indices of $Y^{a\dot{a}}$ and $Z^{\alpha\dot{\alpha}}$ we calculate the action on single particle states $|Y_{a\dot{a}}\rangle \equiv a^\dagger_{a\dot{a}}(p)|0\rangle$ (and similar for $Z_{\alpha\dot{\alpha}}$)

$$
\begin{aligned}
P_{\phi_1}|Y_{1\dot{1}}\rangle &= +|Y_{1\dot{1}}\rangle, & P_{\phi_2}|Y_{1\dot{2}}\rangle &= +|Y_{1\dot{2}}\rangle,\\
P_{\phi_1}|Y_{2\dot{2}}\rangle &= -|Y_{2\dot{2}}\rangle, & P_{\phi_2}|Y_{2\dot{1}}\rangle &= -|Y_{2\dot{1}}\rangle,\\
P_{\psi_1}|Z_{3\dot{3}}\rangle &= +|Z_{3\dot{3}}\rangle, & P_{\psi_2}|Z_{3\dot{4}}\rangle &= +|Z_{3\dot{4}}\rangle,\\
P_{\psi_1}|Z_{4\dot{4}}\rangle &= -|Z_{4\dot{4}}\rangle, & P_{\psi_2}|Z_{4\dot{3}}\rangle &= -|Z_{4\dot{3}}\rangle.
\end{aligned}
\tag{B.7}
$$

Now for the two light-cone charges. The charge $P_-$ after light-cone gauge fixing (see section 2.4) is

$$P_- = \int_0^{R_\sigma} \mathrm{d}\sigma\, p_M \hat{P}_-(x^M) = \int_0^{R_\sigma} \mathrm{d}\sigma\, p_- = R_\sigma,\tag{B.8}$$

as we set $p_- = 1$.

As is expected from the light-cone gauge-fixing procedure (see [41]) and also follows from direct calculation of the YB-deformed T matrices of section 3.1.1, the action of $P_+$ is

$$P_+ = -H_{\text{ws}},\tag{B.9}$$

where $H_{\text{ws}}$ is the worldsheet Hamiltonian. It and the worldsheet momentum are the generators of worldsheet time and space translations respectively. Their action on arbitrary coordinate fields is[20]

$$\hat{H}_{\text{ws}}(x(\tau,\sigma)) = \partial_\tau x(\tau,\sigma), \qquad -\hat{P}_{\text{ws}}(x(\tau,\sigma)) = \partial_\sigma x(\tau,\sigma),\tag{B.10}$$

in position space or

$$\hat{H}_{\text{ws}}(\tilde{x}(\omega,p)) = \mathrm{i}\omega \tilde{x}(\omega,p), \qquad \hat{P}_{\text{ws}}(\tilde{x}(\omega,p)) = \mathrm{i}p \tilde{x}(\omega,p),\tag{B.11}$$

in momentum space. The associated Noether charges are

$$
\begin{aligned}
H_{\text{ws}} &= \int \mathrm{d}\sigma\, p_M \partial_\tau x^M = \int \mathrm{d}p\, \omega_p\, a^\dagger_{M\dot{M}} a^{M\dot{M}},\\
P_{\text{ws}} &= -\int \mathrm{d}\sigma\, p_M \partial_\sigma x^M = \int \mathrm{d}p\, p\, a^\dagger_{M\dot{M}} a^{M\dot{M}},
\end{aligned}
\tag{B.12}
$$

---

[20]We define $P_{\text{ws}}$ with an extra sign to account for the sign difference in our choice of Fourier modes $\mathrm{e}^{\mathrm{i}(\omega\tau - p\sigma)}$. This ensures that $P_{\text{ws}} = \sum_j p_j$.

where again we made use of the on-shell mode expansions of appendix C and hide the $p$ dependence of the quantum operators. Applied to single particle states $|p\rangle = a^\dagger(p)|0\rangle$ of all particles types, this gives

$$H_{\text{ws}}|p\rangle = \omega_p|p\rangle\,, \qquad\qquad P_{\text{ws}}|p\rangle = p|p\rangle\,. \tag{B.13}$$

For the generator $P_+$ this implies the following two relations that are needed in the main text:

$$\hat{P}_+(\tilde{x}) = -\omega\tilde{x}\,, \qquad\qquad P_+|p\rangle = -\omega_p|p\rangle\,. \tag{B.14}$$

# C  Details of the perturbative calculation

To calculate the perturbative T matrix we use standard Feynman diagram methods. We implemented the procedure in Mathematica with the help of the packages *FeynRules* [50] and *FeynArts* [51]. A detailed description of our implementation can be found in app. B of [9]. The data that is different for the calculations of the present paper is the mode expansion, dispersion relation and the kinematical factor.

$r = \Gamma P_\mu \wedge P_\nu$, $P_{\mu,\nu} \neq P_-$  The deformation does not affect the quadratic Lagrangian, only the quartic interaction terms, hence we can use the same data as in the undeformed case. The mode expansion takes the form (where $X^{M\dot{M}}$ stands for $Y^{a\dot{a}}$ and $Z^{\alpha\dot{\alpha}}$)

$$X^{M\dot{M}}(\tau,\sigma) = \frac{1}{4\sqrt{\pi}}\int \frac{\mathrm{d}p}{\sqrt{\omega_p}}\Big(e^{i(p\sigma-\omega_p\tau)}a^{M\dot{M}}(p) + e^{-i(p\sigma-\omega_p\tau)}\epsilon^{MN}\epsilon^{\dot{M}\dot{N}}a^\dagger_{N\dot{N}}(p)\Big),$$
$$\text{with } \omega_p = \sqrt{1+p^2}\,. \tag{C.1}$$

The kinematic factor, that expresses the momentum and energy conservation and reduces the T matrix dependence from four external momenta to two, is also unchanged

$$T(p_1,p_2) = \int \mathrm{d}p_3\,\mathrm{d}p_4\,\delta(p_1+p_2-p_3-p_4)\delta(\omega_1+\omega_2-\omega_3-\omega_4)\,T(p_1,p_2,p_3,p_4)$$
$$= \frac{\omega_1\omega_2}{|p_1\omega_2-p_2\omega_1|}(T(p_1,p_2,p_1,p_2) + T(p_1,p_2,p_2,p_1))\,. \tag{C.2}$$

$r = \Gamma P_- \wedge P_\nu$, $\nu \in \{\phi_1,\phi_2,\psi_1,\psi_2\}$  Here the quadratic Lagrangian gets deformed; to be precise it gets shifted by the expression given in eq. (3.24). This affects the dispersion relation, makes it dependent on the particle species and leads to

$$X^{M\dot{M}}(\tau,\sigma) = \frac{1}{4\sqrt{\pi}}\int \mathrm{d}p \left(\frac{e^{i(p\sigma-\omega_p\tau)}}{\sqrt{\omega_p^{M\dot{M}}}}a^{M\dot{M}}(p) + \frac{e^{-i(p\sigma-\omega_p\tau)}}{\sqrt{\omega_p^{N\dot{N}}}}\epsilon^{MN}\epsilon^{\dot{M}\dot{N}}a^\dagger_{N\dot{N}}(p)\right),$$
$$\text{with } \omega_p^{M\dot{M}} = \omega_0(p-\Gamma P_\nu^{M\dot{M}}) = \sqrt{1+(p-\Gamma P_\nu^{M\dot{M}})^2}\,, \tag{C.3}$$

where $P_\nu^{M\dot{M}}$ are the eigenvalues of $|X^{M\dot{M}}\rangle$ for $P_\nu$, see eq. (B.7). The kinematic factor gets likewise affected by this effective replacement $p \to p-\Gamma P_\nu$:

$$T(p_1,p_2) = \int \mathrm{d}p_3\,\mathrm{d}p_4\,\delta(p_1+p_2-p_3-p_4)\delta(\omega_1+\omega_2-\omega_3-\omega_4)\,T(p_1,p_2,p_3,p_4)$$
$$= \frac{\omega_1\omega_2}{\left|(p_1-\Gamma P_\nu^1)\omega_2-(p_2-\Gamma P_\nu^2)\omega_1\right|}$$
$$\times \big(T(p_1,p_2,p_1-\Gamma(P_\nu^1-P_\nu^3),p_2-\Gamma(P_\nu^2-P_\nu^4))$$
$$+ T(p_1,p_2,p_2-\Gamma(P_\nu^2-P_\nu^3),p_1-\Gamma(P_\nu^1-P_\nu^4))\big)\,. \tag{C.4}$$

The expressions in the last two lines come from the fact that instead of the usual $p_1 = p_3$ and $p_2 = p_4$ we get

$$p_1 - \Gamma P_\nu^1 = p_3 - \Gamma P_\nu^3 \qquad \text{and} \qquad p_2 - \Gamma P_\nu^2 = p_4 - \Gamma P_\nu^4 \tag{C.5}$$

and similarly with $3 \leftrightarrow 4$.

$r = \Gamma P_- \wedge P_+$ Again the Lagrangian gets shifted. This time this results in the replacement $p \to p + \Gamma \omega$. We use the mode expansion

$$X^{M\dot{M}}(\tau, \sigma) = \frac{1}{4\sqrt{\pi}} \int \frac{\mathrm{d}p}{\sqrt{\tilde{\omega}_p}} \Big( e^{i(p\sigma - \omega_p \tau)} a^{M\dot{M}}(p) + e^{-i(p\sigma - \omega_p \tau)} \epsilon^{MN} \epsilon^{\dot{M}\dot{N}} a_{N\dot{N}}^\dagger(p) \Big),$$

$$\text{with} \quad \omega_p = \frac{p\Gamma + \sqrt{1 - \Gamma^2 + p^2}}{1 - \Gamma^2} \quad \text{and} \quad \tilde{\omega}_p = \sqrt{1 - \Gamma^2 + p^2} \,. \tag{C.6}$$

The dispersion $\omega_p$ is the positive energy solution to the shifted relativistic dispersion equation

$$-\omega^2 + (p + \Gamma\omega)^2 = -1 \,. \tag{C.7}$$

The kinematic factor becomes

$$T(p_1, p_2) = \int \mathrm{d}p_3 \, \mathrm{d}p_4 \, \delta(p_1 + p_2 - p_3 - p_4) \delta(\omega_1 + \omega_2 - \omega_3 - \omega_4) \, T(p_1, p_2, p_3, p_4)$$

$$= \frac{\tilde{\omega}_1 \tilde{\omega}_2}{|(p_1 + \Gamma\omega_1)\omega_2 - (p_2 + \Gamma\omega_2)\omega_1|} (T(p_1, p_2, p_1, p_2) + T(p_1, p_2, p_2, p_1)) \,. \tag{C.8}$$

As discussed in section 3.1.2, in both cases that include $P_-$, the interaction Lagrangian, dispersion relation, and kinematic factor all play together nicely to produce the deformed T matrix

$$T^\Gamma(p_1, p_2) = T^0(p_1 - \Gamma P_\nu^1, p_2 - \Gamma P_\nu^2) \,, \tag{C.9}$$

where $T^0$ is the undeformed result of [41, 43].

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
