# Peer review of "Do Drinfeld twists of $AdS_5 \times S^5$ survive light-cone quantization?"

_SciPost Physics Core, doi:SciPost Phys. Core 5, 028 (2022)_

## Round 1 · Referee Report · Anonymous (Referee 1) · 2022-2-24

# 1 Referee Report

This paper contains a very interesting study of the quantisation of models which are abelian Yang-Baxter deformations of the $AdS^5 \times S^5$ superstring theory. For a sub-class of these models an $S$-matrix scheme is setup and the relationship with Drinfeld twists is established, while for another example the authors find obstacles in defining an $S$-matrix due to the impossibility of eliminating the worldsheet coordinate dependence.

The paper is written very clearly. It also contains a sufficient amount of novel material in my opinion to grant its publication. The study is very nice and of definite interest.

I do not really have any request, but rather a set of points which the authors may or may not decide to take on board, or in fact to answer. I do not make the addressing of these points a condition for my recommendation, but I consider them merely as an opportunity for interesting discussion.

1. Above formula (2.6), it is said that "As A and B are operators, the boundary conditions are operator valued." Perhaps it might be helpful to slightly expand on what it means to have operator-valued boundary conditions, on what would otherwise appear to be ordinary fields.

2. Formula (3.18) - it might be helpful to slightly clarify how this is in fact an independent test. Starting from a twisted $S$-matrix one obtains twisted Bethe equations, but that does not naively seem to be an independent test, so another ingredient must be entering at this point.

3. Below (4.17) it is said that "However this completely removes the deformation form the Lagrangian and shifts it into the boundary conditions of the model". I seem to fail to understand why this is portrayed as a negative thing - what would then be the consequent problem with this conclusion?

4. In the Conclusions, it is said that "Interestingly, the resulting S matrix satisfies not the usual, but a shifted version of the (quantum) Yang-Baxter equation." This seems quite a significant issue - perhaps the authors could expand a bit more, especially on the impact this has on integrability.

---

## Round 1 · Referee Report · Anonymous (Referee 2) · 2022-3-15

Report

This paper deals with simple integrable deformations of the AdS5xS5 string of Yang-Baxter type. In particular it considers abelian deformations involving the Cartan directions, which are equivalent to TsT transformations. It is expected that these deformations should be captured by Drinfeld twists of the undeformed S-matrix. The authors verify this by comparing the twisted S-matrix for a BMN string to the tree-level 2-particle S-matrix found from the deformed Lagrangian and showing that they agree. The exception to this is when the deformation involves the light-cone directions used in the gauge fixing. Here the deformation acts instead by a shift in the momenta. They also attempt to analyse the GKP string, but in that case they are not able to find a gauge that allows for a perturbative treatment.

The paper is clearly written and the results are interesting, though not unexpected. The situation for the GKP string is somewhat unsatisfactory, in that it is not clear whether the problem they encounter is simply a technical one that could be easily overcome. Nevertheless, I feel that the paper is interesting and novel enough to warrant publication in SciPost.

---

## Round 2 · List of Changes

We have addressed the points raised in the first report. Regarding 1. we have added a clarifying comment elaborating on the configuration (state) dependence of the boundary conditions in the text above equation (2.6). Regarding 2. we have rephrased the text to make it clear that in this section we were deriving a set of equations to be contrasted against other results later. Regarding 3., we agree that moving the deformation into boundary conditions is not inherently undesirable. However, it simply does not align with our goal to compute S matrices in the deformed model directly. We extended the sentence to clarify our intended meaning. Regarding 4., we agree that this is a subtle and relevant point. We have added a footnote that we believe quantum integrability to be essentially unaffected: the momentum shift in the Lagrangian should not affect the presence of sufficiently many conserved charges, but these charges would pick up particle species dependent momentum shifts, which in turn should lead to the shifted Yang-Baxter equation.
The second referee commented that it is not clear whether our problems surrounding deformations of the GKP string could be easily overcome. We agree that it is desirable to definitively settle this question. However, lack of a resolution is typically not easy to prove. As described, we have attempted various redefinitions and modifications without success. Moreover, the field redefinitions done in the undeformed setting, certainly blur the action of the conserved charges involved in the deformation, on the scattering fields of the Lagrangian. This suggests a hypothetical resolution would require more than e.g. deforming the classical solution we expand around.

---

## Editorial Decision

published